# Stein Bridging: Enabling Mutual Reinforcement between Explicit and Implicit Generative Models

## Abstract

Deep generative models are generally categorized into explicit models and implicit models. The former defines an explicit density form, whose normalizing constant is often unknown; while the latter, including generative adversarial networks (GANs), generates samples without explicitly defining a density function. In spite of substantial recent advances demonstrating the power of the two classes of generative models in many applications, both of them, when used alone, suffer from respective limitations and drawbacks. To mitigate these issues, we propose *Stein Bridging*, a novel joint training framework that connects an explicit (un-normalized) density estimator and an implicit sample generator with Stein discrepancy. We show that the Stein Bridge induces new regularization schemes for both explicit and implicit models. Convergence analysis and extensive experiments demonstrate that the Stein Bridging i) improves the stability and sample quality of the GAN training, and ii) facilitates the density estimator to seek more modes in data and alleviate the mode-collapse issue. Additionally, we discuss several applications of Stein Bridging and useful tricks in practical implementation used in our experiments.

## 1 Introduction

Deep generative model, as a powerful unsupervised framework for learning the distribution of high-dimensional multi-modal data, has been extensively studied in recent literature. Typically, there are two types of generative models (Goodfellow et al., 2014). Explicit models define an explicit (unnormalized) density function, while implicit models learn to sample from the distribution without explicitly defining a density function.

Explicit models have wide applications in undirected graphical models (LeCun et al., 2006; Salakhutdinov & Hinton, 2009; Hinton et al., 2006; Ngiam et al., 2011), random graph theory (Robins et al., 2007), energy-based reinforcement learning (Haarnoja et al., 2017), etc. However, the unknown normalizing constant makes the model hard to train and sample from, and the explicit models might not be able to capture the complex structure of true samples while maintaining tractability. In contrast, implicit models are more flexible in training and easy to sample from, and in pariticular, generative adverarial networks (GANs) have shown great power in learning representations of images, natural languages, graphs, etc. (Goodfellow et al., 2014; Radford et al., 2016; Arjovsky et al., 2017; Brock et al., 2019). Nevertheless, due to the minimax game between generator and discriminator/critic in GANs, the training process often suffers from instability, and produces undesirable samples often associated with missing modes in data or generating extra modes out of data. Therefore, it motivates us to consider jointly learning of two models that can presumably compensate and reinforce each other in the training process. (More discussions about related works are in Appendix A.)

Furthermore, there are many of situations where we do need both an explicit (unnormalized) density and a flexible implicit sampler. For sample evaluation, it is not enough to merely distinguish samples between real and faked one, and one may also expect to provide fine-grained evaluation on generated samples, where the energy values given by the explicit models can be a good metric (Dai et al., 2017). Another situation is outlier detection. Implicit models often leverage all true samples (possibly mixed with corrupted samples) as true examples for training. To make up for the issue, explicit

models could help to detect out-of-distribution samples via the estimated densities (Zhai et al., 2016). Also, when given insufficient observed samples, explicit models may fail to capture an accurate distribution, in which case implicit model may help with data augmentation and facilitate training for density estimation. These situations inspire us to combine both of the worlds so as to take advantage of two models in an effective way.

In this work, we aim at jointly learning explicit and implicit generative models. In our framework, an explicit energy model is used to estimate the unnormalized densities of true samples via minimizing a Stein discrepancy; in the meantime, an implicit generator model is exploited to minimize the Wasserstein metric (or Jensen-Shannon divergence) between distributions of true and generated samples. On top of these, another Stein discrepancy, acting as a bridge between implicit generated samples and explicit estimated densities, is introduced and pushes the two models to achieve a consensus. We show that the Stein bridge allows the two generative models to reinforce each other by imposing new regularizations on both models, which help the generator to output high-quality samples and facilitate the energy model to avoid mode-collapse. Moreover, we show that the joint training helps to stabilize GAN training via a convergence analysis. Extensive experiments on various tasks verify our theoretical findings as well as demonstrate the superiority of proposed methods compared with existing deep energy models and GAN-based models.

## 2 BACKGROUND

In this section, we briefly provide some technical background used in our model.

**Energy Model.** The energy model assigns each data $\mathbf{x} \in \mathbb{R}^d$ with a scalar energy value $E_\phi(\mathbf{x})$, where $E_\phi(\cdot)$ is called the energy function and is parameterized by $\phi$. The model is expected to assign low energy to true samples according to a Gibbs distribution $p_\phi(\mathbf{x}) = \exp\{-E_\phi(\mathbf{x})\}/Z_\phi$, where $Z_\phi$ is a normalizing constant dependent on $\phi$. The normalizing term $Z_\phi$ is often hard to compute, making the training intractable, and various methods are proposed to detour such term (see Appendix A).

**Stein Discrepancy.** Stein discrepancy (Gorham & Mackey, 2015; Liu et al., 2016; Chwialkowski et al., 2016; Oates et al., 2017) is a measure of closeness between two probability distributions that does not require the knowledge for the normalizing constant of one of the compared distributions. Let $\mathbb{P}$ and $\mathbb{Q}$ be two probability distributions on $\mathcal{X} \subset \mathbb{R}^d$, and assume $\mathbb{Q}$ has a (unnormalized) density $q$. The Stein discrepancy $\mathcal{S}(\mathbb{P}, \mathbb{Q})$ is defined as

$$\mathcal{S}(\mathbb{P}, \mathbb{Q}) := \sup_{\mathbf{f} \in \mathcal{F}} \mathbb{E}_{\mathbf{x} \sim \mathbb{P}}[\mathcal{A}_\mathbb{Q} \mathbf{f}(\mathbf{x})] := \sup_{\mathbf{f} \in \mathcal{F}} \{\phi(\mathbb{E}_{\mathbf{x} \sim \mathbb{P}}[\nabla_\mathbf{x} \log q(\mathbf{x}) \mathbf{f}(\mathbf{x})^\top + \nabla_\mathbf{x} \mathbf{f}(\mathbf{x})])\}, \quad (1)$$

where $\mathcal{F}$ is often chosen to be a Stein class (see, e.g., Definition 2.1 in Liu et al. (2016)), $\mathbf{f} : \mathbb{R}^d \to \mathbb{R}^{d'}$ is a vector-valued function called *Stein critic* and $\phi$ is an operation that transforms a $d \times d'$ matrix into a scalar value. One common choice[1] of $\phi$ is trace operation when $d' = d$. If $\mathcal{F}$ is a unit ball in some reproducing kernel Hilbert space (RKHS) with a positive definite kernel $k$, it induces Kernel Stein Discrepancy (KSD). We provide more details in Appendix B.

**Wasserstein Metric.** Wasserstein metric is suitable for measuring distances between two distributions with non-overlapping supports (Arjovsky et al., 2017). The Wasserstein-1 metric between distributions $\mathbb{P}$ and $\mathbb{Q}$ is defined as $\mathcal{W}(\mathbb{P}, \mathbb{Q}) := \min_\gamma \mathbb{E}_{(\mathbf{x}, \mathbf{y}) \sim \gamma}[\|\mathbf{x} - \mathbf{y}\|]$, where the minimization is over all joint distributions with marginals $\mathbb{P}$ and $\mathbb{Q}$. By Kantorovich-Rubinstein duality, it has a dual representation

$$\mathcal{W}(\mathbb{P}, \mathbb{Q}) := \max_D \{\mathbb{E}_{\mathbf{x} \sim \mathbb{P}}[D(\mathbf{x})] - \mathbb{E}_{\mathbf{y} \sim \mathbb{Q}}[D(\mathbf{y})]\}, \quad (2)$$

where the maximization is over all 1-Lipschitz continuous functions.

**Sobolev space and Sobolev dual norm.** Use $L^2$ to denote the canonical Hilbert space on $\mathbb{R}^d$ equipped with an inner product $\langle u, v \rangle_{L^2} := \int_{\mathbb{R}^d} uv d\mathbf{x}$. The Sobolev space $H^1$ is defined as the closure of $C_0^\infty$, the set of smooth functions on $\mathbb{R}^d$ with compact support, with respect to the norm $\|u\|_{H^1} := \left(\int_{\mathbb{R}^d} (u^2 + \|\nabla u\|_2^2) d\mathbf{x}\right)^{1/2}$. For $v \in L^2$, its Sobolev dual norm $\|v\|_{H^{-1}}$ is defined by (Evans & Society, 2010)

$$\|v\|_{H^{-1}} := \sup_{u \in H^1} \left\{\langle v, u \rangle_{L^2} : \int_{\mathbb{R}^d} \|\nabla u\|_2^2 d\mathbf{x} \leq 1, \int_{\mathbb{R}^d} u(\mathbf{x}) d\mathbf{x} = 0\right\}.$$

---

[1]One can also use other forms for $\phi$, like matrix norm when $d' \neq d$ (Liu et al. (2016)).

The constraint $\int_{\mathbb{R}^d} u(\mathbf{x}) d\mathbf{x} = 0$ is necessary in ensuring the finiteness of the supremum. $\|\cdot\|_{H^{-1}}$ can be viewed as a measure of smoothness, which measures the similarity (in terms of largest $L^2$-norm) between $v$ and a subset of smooth functions in $H^1$.

## 3 PROPOSED MODEL

In this section, we formulate our model, *Stein Bridging*, and highlight its regularization effects.

### 3.1 MODEL FORMULATION

We denote by $\mathbb{P}_{\text{real}}$ the underlying real distribution from which the data $\{\mathbf{x}\}$ are sampled. We simultaneously learn two generative models – one explicit and one implicit – that represent estimates of $\mathbb{P}_{\text{real}}$. The explicit generative model has an explicit probability density $\mathbb{P}_E$ proportional to $\exp(-E(\mathbf{x}))$, where $E$ is referred to as an energy function. The implicit generative model transforms an easy-to-sample random noise $\mathbf{z}$ with distribution $P_0$ via a generator $G$ to a generated sample $\widetilde{x} = G(\mathbf{z})$ with distribution $\mathbb{P}_G$. We use the Stein discrepancy as a measure of closeness between the explicit unnormalized density $\mathbb{P}_E$ and the real distribution $\mathbb{P}_{\text{real}}$, and use the Wasserstein metric as a measure of closeness between the implicit distribution $\mathbb{P}_G$ and $\mathbb{P}_{\text{real}}$.

To jointly learn the two generative models $\mathbb{P}_G$ and $\mathbb{P}_E$, arguably the most straightforward approach is to minimize the sum of the Stein discrepancy and the Wasserstein metric:

$$\min_{E,G} \mathcal{W}(\mathbb{P}_{\text{real}}, \mathbb{P}_G) + \lambda \mathcal{S}(\mathbb{P}_{\text{real}}, \mathbb{P}_E),$$

where $\lambda \geq 0$ is a weight coefficient. However, this approach appears no different than learning the two generative models separately. To better train the model, we incorporate the objective another term $\mathcal{S}(\mathbb{P}_G, \mathbb{P}_E)$ – called the *Stein bridge* – that measures the closeness between the explicit unnormalized density $\mathbb{P}_E$ and the implicit distribution $\mathbb{P}_G$:

$$\min_{E,G} \mathcal{W}(\mathbb{P}_{\text{real}}, \mathbb{P}_G) + \lambda_1 \mathcal{S}(\mathbb{P}_{\text{real}}, \mathbb{P}_E) + \lambda_2 \mathcal{S}(\mathbb{P}_G, \mathbb{P}_E), \tag{3}$$

where $\lambda_1, \lambda_2 \geq 0$ are weight coefficients. Although the Stein bridge might seem redundant mathematically, we show that it helps regularize the models in Section 3.2.

The Wasserstein term in (3) is implemented using its equivalent dual representation (2). The two Stein terms in (3) can be implemented using (1) with either a Stein critic parameterized by a neural network, or the Kernel Stein Discrepancy. To reduce the computational cost, the two Stein critics share their parameters, namely, kernels or neural networks. A scheme of our framework is presented in Fig. 1. We also discuss some related works that attempt to combine both of the worlds (such as energy-based GAN, contrastive learning and cooperative learning) in Appendix A.3, and highlight the difference between our method and theirs in terms of the objective in Table 1.

**Remark**. In general, we can also choose other statistical distances in (3) to measure closeness between probability distributions. For example, the Wasserstein metric $\mathcal{W}(\mathbb{P}_{\text{real}}, \mathbb{P}_G)$ can be replaced by other common choices for implicit generative models, such as Jensen-Shannon divergence used in the original GAN paper (Goodfellow et al., 2014). If the normalizing constant of $\mathbb{P}_E$ is known or easy to calculate, one can replace the Stein discrepancy by the Kullback-Leibler divergence, which is equivalent to the maximum likelihood estimation. We present details for model specifications in various forms and training algorithm in Appendix E.2.

### 3.2 REGULARIZATION EFFECTS BY VIRTUE OF THE STEIN BRIDGE

The intuitive motivation of the Stein bridge term in (3) is to push the two models to achieve a consensus. In this subsection, we theoretically show that the Stein bridge allows the two models to reinforce each other by imposing regularizations on the critics.

#### 3.2.1 KERNEL SOBOLEV DUAL NORM REGULARIZATION ON THE WASSERSTEIN CRITIC

We show the regularization effect of the Stein bridge on the Wasserstein critic. Define the *kernel Sobolev dual norm* as

$$\|D\|_{H^{-1}(\mathbb{P};k)} := \sup_{u \in C_0^\infty} \left\{ \langle D, u \rangle_{L^2(\mathbb{P})} : \mathbb{E}_{\mathbf{x}, \mathbf{x}' \sim \mathbb{P}}[\nabla u(\mathbf{x})^\top k(\mathbf{x}, \mathbf{x}') \nabla u(\mathbf{x}')] \leq 1, \ \mathbb{E}_{\mathbb{P}}[h] = 0 \right\}.$$

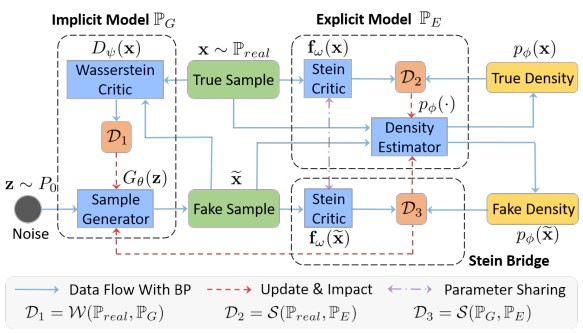

| Model | Objective |
|---|---|
| GAN | $\mathcal{D}_1$ |
| Energy Model | $\mathcal{D}_2$ |
| Energy-based GAN (Zhao et al. (2017)) | $\mathcal{D}_1$ |
| Contrastive Learning (Kim & Bengio (2017)) | $\mathcal{D}_2$ |
| Cooperative Learning (Xie et al. (2018)) | $\mathcal{D}_2 + \mathcal{D}_3$ |
| Stein Bridging (ours) | $\mathcal{D}_1 + \mathcal{D}_2 + \mathcal{D}_3$ |

Table 1: Comparison of objectives between different generative models, where $\mathcal{D}_1 := \mathcal{D}_1(\mathbb{P}_{\text{real}}, \mathbb{P}_G)$, $\mathcal{D}_2 := \mathcal{D}_2(\mathbb{P}_{\text{real}}, \mathbb{P}_E)$ and $\mathcal{D}_3 := \mathcal{D}_3(\mathbb{P}_G, \mathbb{P}_E)$ denote general statistical distances between two distributions.

Figure 1: Model framework for *Stein Bridging* which jointly train an implicit sample generator and an explicit (unnormalized) density estimator via a Stein bridge.

which can be viewed as a kernel generalization of the Sobolev dual norm defined in Section 2, which reduces to the Sobolev dual norm when $k(\mathbf{x}, \mathbf{x}') = \mathbb{I}(\mathbf{x} = \mathbf{x}')$ and $\mathbb{P}$ being the Lebesgue measure. We have the following result.

**Theorem 1.** *Assume that $\{\mathbb{P}_G\}_G$ exhausts all continuous probability distributions and $\mathcal{S}$ is chosen as kernel Stein discrepancy. Then problem* (3) *is equivalent to*

$$\min_E \max_D \left\{ \mathbb{E}_{\mathbf{y} \sim \mathbb{P}_E}[D(\mathbf{y})] - \mathbb{E}_{\mathbf{x} \sim \mathbb{P}_{\text{real}}}[D(\mathbf{x})] - \frac{1}{4\lambda_2} \|D\|_{H^{-1}(\mathbb{P}_E; k)} \right\}.$$

According to Section 2, the regularization term would penalize the non-smoothness of the Wasserstein critic $D$, which is in the same spirit of gradient-based penalty (e.g., Gulrajani et al. (2017); Roth et al. (2017)), but with a new way to encouraging smoothness.

Another way to interpret the Sobolev dual norm penalty is by observing that if $k(\mathbf{x}, \mathbf{x}') = \mathbb{I}(\mathbf{x} = \mathbf{x}')$ and and $\mathbb{E}_{\mathbb{P}_E}[D] = 0$ (Villani, 2008), then

$$\|D\|_{H^{-1}(\mathbb{P}_E; k)} = \lim_{\epsilon \to 0} \frac{\mathcal{W}_2((1 + \epsilon D)\mathbb{P}_E, \mathbb{P}_E)}{\epsilon},$$

where $\mathcal{W}_2$ denotes the 2-Wasserstein metric. Therefore, the regularization ensures that $D$ would not change suddenly on the high-density region of $\mathbb{P}_E$, and the explicit model reinforces the learning of the Wasserstein critic.

### 3.2.2 LIPSCHITZ REGULARIZATION ON THE STEIN CRITIC

We next investigate how the Stein bridge helps to regularize the Stein critic. Recall that the two Stein terms in (3) share the same Stein critic. We have the following result.

**Theorem 2.** *Assume $\{\mathbb{P}_G\}_G$ exhausts all continuous probability distributions, and the Stein class defining the Stein discrepancy is compact (in some linear topological space). Then problem* (3) *is equivalent to*

$$\min_E \max_{\mathbf{f}} \{\lambda_1 \mathcal{S}(\mathbb{P}_{\text{real}}, \mathbb{P}_E) + \lambda_2 \mathbb{E}_{\mathbf{x} \sim \mathbb{P}_{\text{real}}}[M_{\lambda_2 \mathcal{A}_{\mathbb{P}_E} \mathbf{f}}(\mathbf{x})]\},$$

*where $M_{\lambda_2 \mathcal{A}_{\mathbb{P}_E} \mathbf{f}}(\cdot)$ denotes the (generalized) Moreau-Yosida regularization of the function $\mathcal{A}_{\mathbb{P}_E} \mathbf{f}$ with parameter $\lambda_2$, i.e., $M_{\lambda_2 \mathcal{A}_{\mathbb{P}_E} \mathbf{f}}(\mathbf{x}) = \min_{\mathbf{y} \in \mathcal{X}} \{\mathcal{A}_{\mathbb{P}_E} \mathbf{f}(\mathbf{y}) + \frac{1}{\lambda_2} \|\mathbf{x} - \mathbf{y}\|\}$.*

Theorem 2 shows that the Stein bridge, together with the Wasserstein metric $\mathcal{W}(\mathbb{P}_{\text{real}}, \mathbb{P}_G)$, plays as a smoothness regularization on the Stein critic $\mathbf{f}$ via Moreau-Yosida regularization, which smoothens the Stein operator $\mathcal{A}_{\mathbb{P}_E} \mathbf{f}$ and further encourages the energy model to seek more modes in data instead of focusing on some dominated modes, thus helps alleviate the mode-collapse issue. To the best of our knowledge, this suggests a novel regularization scheme for Stein-based GAN.

## 4 CONVERGENCE ANALYSIS

In Section 3.2, we justify Stein Bridging by showing the regularization effects. In this section, we further show that it could help to stabilize GAN training with local convergence guarantee. To this end, we first compare the behaviors of WGAN, likelihood- and entropy-regularized WGAN, and our Stein Bridging under SGD via an easy to comprehend toy example. Then we give a formal result that interprets why the introduction of (unnormalized) density estimator could stablize GAN training and help for convergence.

### 4.1 ANALYSIS OF A LINEAR SYSTEM

The training for minimax game in GAN is difficult. When using traditional gradient methods, the training would suffer from some oscillatory behaviors (Goodfellow (2017); Liang & Stokes (2019)). In order to better understand the optimization behaviors, we first study a one-dimension linear system that provides some insights on this problem. Note that such toy example (or a similar one) is also utilized by Gidel et al. (2019); Nagarajan & Kolter (2017) to shed lights on the instability of WGAN training[2]. Consider a linear critic $D_\psi(x) = \psi x$ and generator $G_\theta(z) = \theta x$. Then the Wasserstein GAN objective can be written as a constrained bilinear problem: $\min_\theta \max_{|\psi| \le 1} \psi \mathbb{E}[x] - \psi\theta\mathbb{E}[z]$, which could be further simplified as an unconstrained version (the behaviors could be generalized to multi-dimensional cases (Gidel et al. (2019))):

$$\min_\theta \max_\psi \psi - \psi \cdot \theta. \tag{4}$$

Unfortunately, such simple objective cannot guarantee convergence by traditional gradient methods like SGD with alternate updating[3]: $\theta_{k+1} = \theta_k + \eta\psi_k,, \psi_{k+1} = \psi_k + \eta(1 - \theta_{k+1})$. Such optimization would suffer from an oscillatory behavior, i.e., the updated parameters go around the optimum point ($[\psi^*, \theta^*] = [0, 1]$) forming a circle without converging to the centrality, which is shown in Fig. 2(a). A recent study in Liang & Stokes (2019) theoretically show that such oscillation is due to the interaction term in (4).

One solution to the instability of GAN training is to add (likelihood) regularization, which has been widely studied by recent literatures (Warde-Farley & Bengio (2017); Li & Turner (2018)). With regularization term, the objective changes into $\min_\theta \max_{|\psi| \le 1} \psi \mathbb{E}[x] - \psi\theta\mathbb{E}[z] - \lambda\mathbb{E}[\log \mu(\theta z)]$, where $\mu(\cdot)$ denotes the likelihood function and $\lambda$ is a hyperparameter. A recent study (Tao et al. (2019)) proves that when $\lambda < 0$ (likelihood-regularization), the extra term is equivalent to maximizing sample evidence, helping to stabilize GAN training; when $\lambda > 0$ (entropy-regularization), the extra term maximizes sample entropy, which encourages diversity of generator. Here we consider a Gaussian likelihood function for generated sample $x'$, $\mu(x') = \exp(-\frac{1}{2}(x' - b)^2)$ which is up to a constant, and then the objective becomes (see Appendix D.1 for details):

$$\min_\theta \max_\psi \psi - \psi \cdot \theta - \lambda(\theta^2 - \theta). \tag{5}$$

The above system would converge with $\lambda < 0$ and diverge with $\lambda > 0$ in gradient-based optimization, shown in Fig. 2(a). Another issue of likelihood-regularization is that the extra term changes the optimum point and makes the model converge to a biased distribution, as proved by Tao et al. (2019). In this case, one can verify that the optimum point becomes $[\psi^*, \theta^*] = [-\lambda, 1]$, resulting a bias. To avoid this issue, Tao et al. (2019) proposes to temporally decrease $|\lambda|$ through training. How-

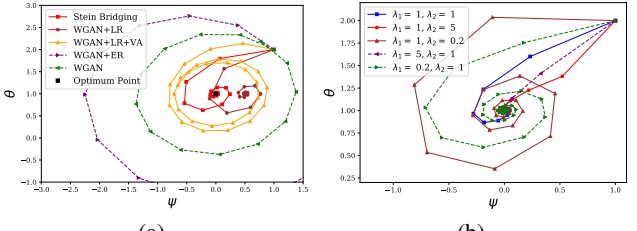

(a)                                   (b)

Figure 2: (a) Numerical iterations for SGD training of WGAN, likelihood-regularized WGAN (WGAN+LR), variational annealing for WGAN+LR (WGAN+LR+VA), entropy-regularized WGAN (WGAN+ER) and our Stein Bridging. (b) Stein Bridging with different $\lambda_1$ and $\lambda_2$.

ever, such method would also be stuck in oscillation when $|\lambda|$ gets close to zero as is shown in Fig. 2(a).

---

[2]Our theoretical discussions focus on WGAN, and we also compare with original GAN in the experiments.

[3]Here, we adopt the most widely used alternate updating strategy. The simultaneous updating, i.e., $\theta_{k+1} = \theta_k + \eta\psi_k$ and $\psi_{k+1} = \psi_k + \eta(1 - \theta_k)$, would diverge in this case.

Finally, let us consider our proposed model. We also simplify the density estimator as a basic energy model $p_\phi(x) = \exp(-\frac{1}{2}x^2 - \phi x)$ whose score function is $\nabla_x \log p_\phi(x) = -x - \phi$. Then if we specify the two Stein discrepancies in (3) as KSD, we have the objective (see Appendix D.1 for details),

$$\min_\theta \max_\psi \min_\phi \psi - \psi \cdot \theta + \frac{\lambda_1}{2}(1 + \phi)^2 + \frac{\lambda_2}{2}(\theta + \phi)^2. \tag{6}$$

Interestingly, one can verify that for $\forall \lambda_1, \lambda_2$, the optimum point remains the same $[\psi^*, \theta^*, \phi^*] = [0, 1, -1]$. Then we show that the optimization can guarantee convergence to $[\psi^*, \theta^*, \phi^*]$.

**Proposition 1.** *Using alternate SGD for (6) geometrically decreases the square norm* $N_t = |\psi^t|^2 + |\theta - 1|^2 + |\phi + 1|^2$, *for any* $0 < \eta < 1$ *with* $\lambda_1 = \lambda_2 = 1$,

$$N_{t+1} = (1 - \eta^2(1 - \eta)^2)N_t. \tag{7}$$

In Fig. 2(a), we can see that Stein Bridging achieves a good convergence to the right optimum. Compared with (4), the objective (6) adds a new bilinear term $\phi \cdot \theta$, which acts like a connection between the two generator and estimator, and two other quadratic terms, which help to push the values to decrease through training. The added terms and the original terms in (6) cooperate to guarantee convergence to a unique optimum. (More discussions in Appendix D.1).

We further generalize the analysis to multi-dimensional bilinear system $F(\psi, \theta) = \theta^\top \mathbf{A} \psi - \mathbf{b}^\top \theta - \mathbf{c}^\top \psi$ which is extensively used by researches for analysis of GAN stability (Goodfellow (2017); Gemp & Mahadevan (2018); Liang & Stokes (2019); Gidel et al. (2019)). For any bilinear system, with the added term $H(\phi, \theta) = \frac{1}{2}(\theta + \phi)^\top \mathbf{B}(\theta + \phi)$ where $\mathbf{B} = (\mathbf{A}\mathbf{A}^\top)^{\frac{1}{2}}$ to the objective, we can prove that i) the optimum point remains the same as the original system (Proposition 2) and ii) using alternate SGD algorithm for the new objective can guarantee convergence (Theorem 4). The results are given in Appendix D.3.

## 4.2 Local Convergence for a General Model

To study the convergence for Stein Bridging, we proceed to consider a general optimization objective

$$\min_\theta \max_\psi \min_\phi L(\theta, \psi, \phi),$$

where $L(\theta, \psi, \phi) = F(\theta, \psi) + H(\theta, \phi)$, and $\omega_f = [\theta, \psi]$ and $\omega_h = [\theta, \phi]$ ($\theta$ is a shared parameter set). Use $\omega^* = [\theta^*, \psi^*, \phi^*]$ to denote the optimum point of $L$ and $\omega_f^* = [\theta^*, \psi^*]$, $\omega_h^* = [\theta^*, \phi^*]$ represent the optimum points of $F$ and $H$ respectively. Define $\Omega_f = \Omega_\theta \times \Omega_\psi$ and $\Omega_h = \Omega_\theta \times \Omega_\phi$, where $\Omega_\theta, \Omega_\psi, \Omega_\phi$ denote constraint sets for $\theta, \psi, \phi$ respectively. Function $H$ is $\mu$-strongly convex, and $F$ is $\mu$-strongly convex for $\theta$ and $\mu$-strongly concave for $\psi$ (see Appendix E.4 for definition of strongly convex condition). Here we define $h(\omega_h) = \nabla_\theta H + \nabla_\phi H$, $f(\omega_f) = \nabla_\theta F - \nabla_\psi F$, and then we have the following theorem.

**Theorem 3.** *If $F$ is $\mu$-strongly convex-concave and $H$ is $\mu$-strongly convex, we can leverage the alternate SGD algorithm, i.e.*

$$\omega_h^{t+1} = P_{\Omega_h}(\omega_h^{t+1/2} - \eta h(\omega_h^{t+1/2})), \tag{8}$$

$$\omega_f^{t+1} = P_{\Omega_f}(\omega_f^{t+1/2} - \eta f(\omega_f^{t+1/2})), \tag{9}$$

*where $\omega_h^{t+1/2} = [\theta^t, \psi^t]$, $\omega_h^{t+1} = [\theta^{t+1/2}, \psi^{t+1}]$, $\omega_f^{t+1/2} = [\theta^{t+1/2}, \phi^t]$, $\omega_f^{t+1} = [\theta^{t+1}, \phi^{t+1}]$, and $P_\Omega(\omega) = \arg\min_{\omega' \in \Omega} \|\omega - \omega'\|_2^2$ denotes the projection mapping to $\Omega$. Then we can achieve the convergence by using $\frac{1}{2\mu} < \eta < \frac{1}{\mu}$.*

Theorem 3 shows that Stein Bridging could converge to at least a local optimum. Due to the unknown and intricate landscape of deep neural networks, the global optimization and convergence analysis for GAN has remained as an unexplored problem. Despite the fact that strong convexity assumption cannot be guaranteed with deep neural networks, the optimization could converge to a stable point once there exists a local region that satisfies the strongly convex conditions. In the experiments, we will empirically compare the training stability of each method on various datasets to validate our theoretical discussions.

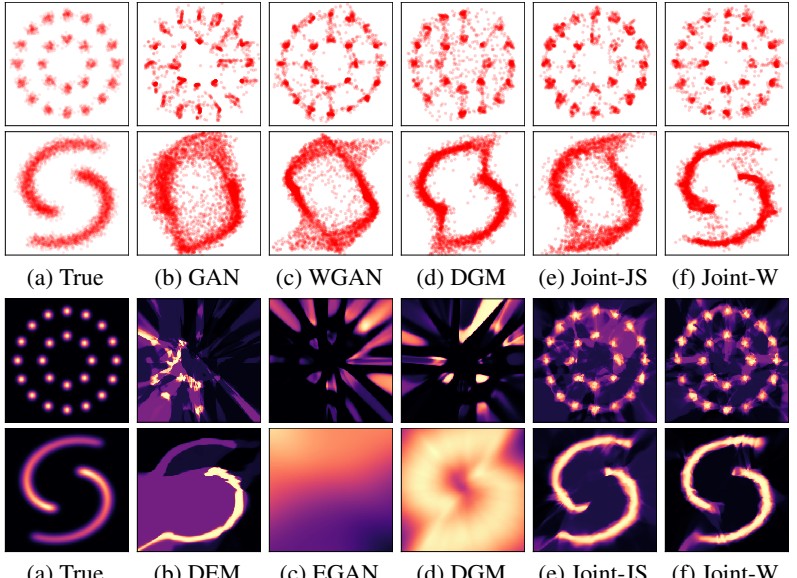

Figure 3: (a) True samples and (b)∼(f) generated samples produced by the generators of different methods on Two-Circle (upper line) and Two-Spiral (bottom line) datasets.

(a) True    (b) GAN    (c) WGAN    (d) DGM    (e) Joint-JS    (f) Joint-W

Figure 4: (a) True densities and (b)∼(f) estimated densities given by the estimators of different methods on Two-Circle (upper line) and Two-Spiral (bottom line) datasets.

(a) True    (b) DEM    (c) EGAN    (d) DGM    (e) Joint-JS    (f) Joint-W

## 5 EXPERIMENTS

In this section, we conduct experiments to verify the effectiveness of proposed method from multi-faceted views. First, we select three tasks with different evaluation metrics in Section 5.1, 5.2 and 5.3. Then we further discuss some applications of joint training as well as some useful tricks in Section 5.4, 5.5 and 5.6. The codes will be released later.

We consider two synthetic datasets with mixtures of Gaussian distributions: Two-Circle and Two-Spiral. The first one is composed of 24 Gaussian mixtures that lie in two circles. Such dataset is extended from the 8-Gaussian-mixture scenario which is widely used in previous GAN papers and is more difficult, so that we can use it to test the quality of generated samples and mode coverage of learned energy. The second synthetic dataset consists of 100 Gaussian mixtures whose centers are densely arranged on two centrally symmetrical spiral-shaped curves. This dataset can be used to examine the power of generative model on complicated data distributions. The ground-truth distributions and samples are shown in Fig. 3 (a) and Fig. 4 (a). Furthermore, we also apply the methods to MNIST and CIFAR datasets which require the model to deal with high-dimensional data. In each dataset, we use observed samples as input of the model and leverage them to train the generators and the estimators. The details for each dataset are reported in Appendix E.1.

In our experiments, we also replace the Wasserstein metric in (3) by JS divergence. To well distinguish different specifications, we term the model Joint-W if using Wasserstein metric and Joint-JS if using JS divergence in this section. We consider several competitors. First, for implicit generative models, we consider valina GAN, WGAN-GP (Gulrajani et al. (2017)), likelihood-regularized GAN/WGAN-GP (short as GAN+LR/WGAN+LR), entropy-regularized GAN/WGAN-GP (short as GAN+ER/WGAN+ER) and a recently proposed variational annealing regularization (Tao et al. (2019)) for GAN (short as GAN+VA/WGAN+VA) to compare the quality of generated samples. We employ the denoising auto-encoder to estimate the gradient for regularization penalty, which is proposed by Alain & Bengio (2014) and utilized by Tao et al. (2019). Second, for explicit models, we consider Deep Energy Model (DEM) which is optimized based on Stein discrepancy, and energy-based GAN (EGAN) (Dai et al. (2017)). Besides, we also compare with Deep Directed Generative (DGM) Model (Kim & Bengio (2017)) which adopts contrastive divergence to unite sample generator and energy estimator. See Appendix A for brief introduction of these methods and Appendix E.3 for implementation details for each method.

### 5.1 SAMPLE QUALITY OF IMPLICIT MODEL

Calibrating explicit (unnormalized) density model with implicit generator is expected to improve the quality of generated samples. In Fig. 3 and Fig. 4 we show the results of different generators in Two-Circle and Two-Spiral datasets. As we can see, in Two-Circle, there are a large number of gen-

| MNIST (Conditional) | | | MNIST (Unconditional) | | | CIFAR-10 (Unconditional) | | |
|---|---|---|---|---|---|---|---|---|
| Method | Score | CEPC | Method | Score | CEPC | Method | Score | CEPC |
| DCGAN | 8.43 | 0.168 | WGAN-GP | 7.71 | 0.256 | WGAN-GP | 6.80 | 0.153 |
| DCGAN+LR | 8.40 | 0.171 | WGAN+LR | 7.82 | 0.243 | WGAN+LR | 6.89 | 0.154 |
| DCGAN+ER | 8.33 | 0.179 | WGAN+ER | 7.75 | 0.252 | WGAN+ER | 6.99 | 0.156 |
| DCGAN+VA | 8.40 | 0.172 | WGAN+VA | 7.74 | 0.254 | WGAN+VA | 6.95† | 0.154 |
| DGM | 8.15 | 0.201 | DGM | 6.87 | 0.372 | DGM | 4.79 | 0.146 |
| Joint-JS(ours) | **8.53** | **0.156** | Joint-W(ours) | **7.90** | **0.231** | Joint-W(ours) | **7.11** | **0.151** |

Table 2: Inception scores (higher is better) and conditional entropies (short as CEPC and lower is better) on MNIST and CIFAR-10. † We directly use the best result reported in their paper.

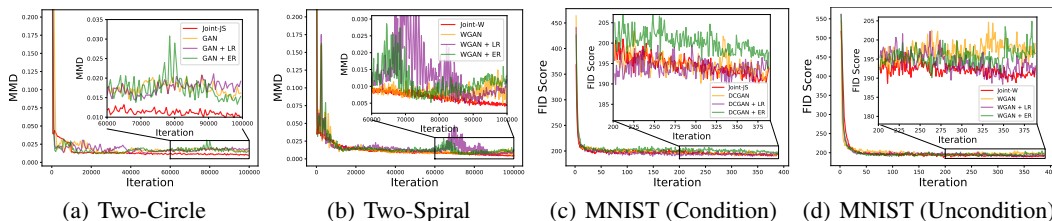

(a) Two-Circle      (b) Two-Spiral      (c) MNIST (Condition)      (d) MNIST (Uncondition)

Figure 5: Learning curves of Joint-W (resp. Joint-JS) compared with WGAN (resp. GAN or DC-GAN) and its regularization-based variants.

erated samples given by GAN, WGAN-GP and DGM (the worst one in this case) locating between two Gaussian components, and the boundary for each component is not distinguishable. Since the ground-truth densities of regions between two components are very low, such generated samples possess low-quality, which depicts that these models capture the combinations of two dominated features (i.e., modes) in the data but such combination does not make sense in practice. By contrast, Joint-JS and Joint-W could alleviate such issue, reduce the low-quality samples and produce more distinguishable boundaries for components. In Two-Spiral, similarly, the generated samples given by GAN and WGAN-GP form a circle instead of two spirals while the samples of DGM 'link' two spirals. Joint-JS manages to focus more on true high densities compared to GAN and Joint-W provides the best results. To quantitatively measure the sample quality, we adopt two metrics: Maximum Mean Discrepancy (MMD) and High-quality Sample Rate (HSR). The detailed definitions are given in Appendix E.4 and we report the results in Table 5.

We visualize the generated digits/images on MNIST/CIFAR-10 datasets in Fig. 9 and Fig. 10 and use Inception Score and conditional entropy of predicted classes (CEPC) to measure the sample quality (See Appendix E.4 for details). As shown in Table 2, Joint-W (resp. Joint-JS) is superior than WGAN-GP (resp. DCGAN), regularized WGAN (resp. DCGAN) and DGM. The CEPC characterizes how well the picture can be distinguished by a pre-trained classifier, i.e., the quality of picture, so the results depict that proposed method could give higher-quality generated pictures.

## 5.2 DENSITY ESTIMATION OF EXPLICIT MODEL

Another advantage of joint learning is that the generator could help the density estimator to capture more accurate distribution. As shown in Fig 3, both Joint-JS and Joint-W manage to capture all Gaussian components while other methods miss some of modes. In Fig 4, Joint-JS and Joint-W exactly fit the ground-truth distribution. By contrast, DEM misses one spiral while EGAN degrades to a uniform-like distribution. DGM manages to fit two spirals but allocate high densities to regions that have low densities in the groung-truth distribution. To quantitatively measure the performance, we introduce three evaluation metrics: KL & JS divergence between the ground-truth and estimated densities and Area Under the Curve (AUC) for false-positive rate v.s. true-positive rate where we select points with true high (resp. low) densities as positive (resp. negative) examples. The detailed information and results are given in Appendix E.4 and Table 5 respectively. The values show that Joint-W and Joint-JS could provide more accurate (unnormalized) density estimation than other competitors.

We also rank the generated digits (and true digits) on MNIST w.r.t the densities given by the energy model in Fig. 11, Fig. 12 and Fig. 13. As depicted in the figures, the digits with high densities (or

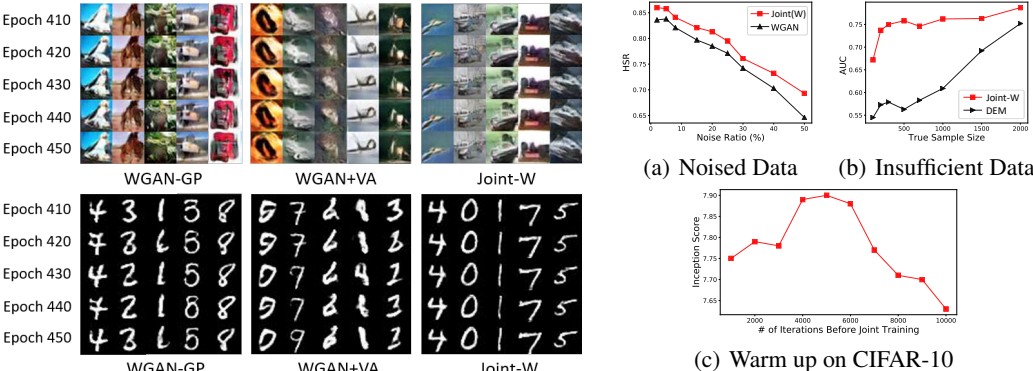

Figure 6: Generated digits (resp. images) given by the same noise $z$ in adjacent training epochs on MNIST (reps. CIFAR) dataset.

Figure 7: Joint-W with (a) noised data, (b) insufficient data and (c) 'warm up' iterations before joint training.

low densities) given by Joint-JS possess enough diversity (the thickness, the inclination angles as well as the shapes of digits diverses). By constrast, all the digits with high densities given by DGM tend to be thin and digits with low densities are very thick. Also, as for EGAN, digits with high (or low) densities appear to have the same inclination angle (for high densities, '1' keeps straight and '9' 'leans' to the left while for low densities, just the opposite). Such phenomenon indicates that DGM and EGAN tend to allocate high (or low) densities to data with certain modes and would miss some modes that possibly possess high densities in ground-truth distributions. Fortunately, our method overcomes the issue and manages to capture complicated distributions.

## 5.3 ENHANCING THE STABILITY OF GAN

Our discussions and analysis show that joint training helps to stabilize GAN training. In Fig. 5 we present the learning curves of Joint-W (resp. Joint-JS) compared with WGAN (resp. GAN or DCGAN) and its regularization-based variants on different datasets. One can clearly see from the curves that joint training could reduce the variance of metric values especially during the second half of training. Furthermore, we visualize the generated pictures given by the same noise $z$ in adjacent epochs in Fig. 6. The results show that Joint-W outputs more stable generation in adjacent epochs while the generated samples given by WGAN-GP and WGAN+VA exhibit an obvious variation. Especially, some digits generated by WGAN-GP and WGAN+VA change from one class to another. Such phenomenon is quite similar to the oscillatory behavior with non-convergence in optimization that we discuss in Section 4.1.

Another issue discussed in Section 4.1 is the bias of model distribution for regularized GAN methods. To quantify this evaluation, we calculate $l_1$ and $l_2$ distances between the means of 50000 generated digits (resp. images) and 50000 true digits (resp. images) in MNIST (reps. CIFAR-10). The results are shown in Table 3. The smaller distances given by Joint-W indicate that it converges to a better local optimum with smaller bias from the original data distribution. Also, in Table 6 (resp. Table 7), we report the distances for digits (resp. images) in each class on MNIST (resp. CIFAR).

## 5.4 DETECTING OUT-OF-DISTRIBUTION SAMPLES

The explicit model estimates densities for each sample and one of its applications is to detect outliers in the input data. Here, we adopt CIFAR-10 to measure the ability of our estimator to distinguish the in-distribution samples and (true/false) out-of-distribution samples. We consider four situations and in each case, we consider the test images of CIFAR-10 as positive set (expected to allocate high densities) and construct a negative set (expected to allocate low densities). We let the model output densities for images in two sets, rank them according to the densities and plot the ROC curve for false-positive rate v.s. true-positive rate in Fig. 8. In the first case, we flip each image in the positive set as negative set. Note that such flipped images are not out-of-distribution samples, so the model is expected to allocate high densities to them, i.e., the ROC curve should be close to a straight line from $(0, 0)$ to $(1, 1)$. The results show that Joint-W, EGAN and DEM give the exact results while DGM assigns all flipped images with lower densities, which means that it fails to capture the semantics

|        | MNIST | | CIFAR | |
| Method | $l_1$ Dis | $l_2$ Dis | $l_1$ Dis | $l_2$ Dis |
| --- | --- | --- | --- | --- |
| WGAN-GP | 13.80 | 0.93 | 80.98 | 1.72 |
| WGAN+LR | 12.91 | 0.86 | 82.96 | 1.81 |
| WGAN+ER | 12.26 | 0.77 | 72.28 | 1.59 |
| WGAN+VA | 12.38 | 0.78 | 69.01 | 1.53 |
| DGM | 12.12 | 0.79 | 179.30 | 3.95 |
| Joint-W | **11.82** | **0.73** | **64.23** | **1.41** |

Table 3: Distances between means of generated digits (resp. images) and ground-truth digits (resp. images) on MNIST (resp. CIFAR-10).

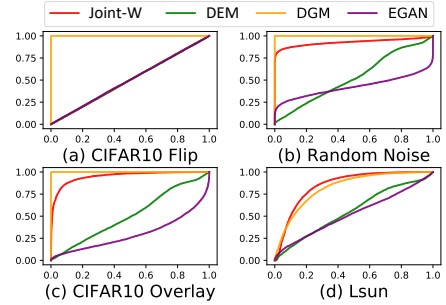

Figure 8: ROC curves for evaluation of outlier detection on CIFAR-10.

in images. In the following three cases, we i) generate random noise, ii) average two images with different CIFAR classes, and iii) adopt Lsun Bedroom dataset as the negative set, respectively. In these situations, the model is expected to distinguish the images in two sets. The results in Fig. 8 show that DGM provides the best results while the performance of Joint-W is quite close to DGM and much better than DEM and EGAN.

## 5.5 Addressing Data Insufficiency and Noisy Data

We proceed to test the model performance in some extreme situations where the observed samples are mixed with noises or the observed samples are quite insufficient. The results are presented in Fig. 7(a) where we add different ratios of random noise to the true samples in Two-Circle dataset and Fig. 7(b) where we only sample insufficient data for training in Two-Spiral dataset. The details are in Appendix E.1. The noise in data impacts the performance of WGAN and Joint-W, but comparatively, the performance decline for Joint-W is less insignificant than WGAN, which indicates better robustness of joint training w.r.t noised data. In Fig. 7(b), when the sample size decreases from 2000 to 100, the AUC value of DEM declines dramatically, showing its dependency on sufficient training samples. By contrast, the AUC of Joint-W exhibits a small decline when the sample size is more than 500 and suffers from an obvious decline when it is less than 300. Such phenomenon demonstrates lower sensitivity of joint training to observed sample size.

## 5.6 When to Start Joint Learning

In our experiment, we also observe an interesting phenomenon: the performance achieved at convergence would be better if we start joint training after some iterations with independent training for the generator and the estimator. In other words, at the beginning, we could set $\lambda_2 = 0$ (or some very small values) in (3) and after some iterations set it as a normal level. We report the inception scores on MNIST with different numbers of iterations for independent training in Fig. 7(c) where we can see that the score firstly goes up and then goes down when we increase iterations for independent training. Such phenomenon is quite similar to the 'warm up' trick used for training deep networks where one can use small learning rates at iterations in the begining and amplify its value for further training. One intuitive reason behind this phenomenon is that at the beginning, both the generator and estimator are weak and if we minimize the discrepancy between them at this point, they would possibly constrain each other and get limited in some bad local optima. When they become strong enough after some training iterations, uniting them through joint training would help them compensate and reinforce each other as our discussions.

## 6 Conclusions

In this paper, we aim at uniting the training for implicit generative model (represented by GAN) and explicit generative model (represented by a deep energy-based model). Besides two loss terms for GAN and energy-based model, we introduce the third loss characterized via Stein discrepancy between the generator in GAN and the energy-based model. Theoretically, we show that joint training could i) help to stablize GAN training and facilitate its convergence, and ii) enforcing dual regularization effects on both models and help to escape from local optima in optimization. We also conduct extensive experiments with different tasks and application senarios to verify our theoretical findings as well as demonstrate the superiority of our method compared with various GAN models and deep energy-based models.

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

# A  LITERATURE REVIEWS

We discuss some of related literatures and shed lights on the relationship between our work with others.

## A.1  EXPLICIT GENERATIVE MODELS

Explicit generative models are interested in fitting each instance with a scaler (unnormalized) density expected to explicitly capture the distribution behind data. Such densities are often up to a constant and called as energy functions which are common in undirected graphical models (LeCun et al. (2006)). Hence, explicit generative models are also termed as energy-based models. An early version of energy-based models is the FRAME (Filters, Random field, And Maximum Entropy) model (Zhu et al. (1997); Wu et al. (2000)). Later on, some works leverage deep neural networks to model the energy function (Ngiam et al. (2011); Xie et al. (2016b)) and pave the way for researches on deep energy model (DEM) (e.g., Liu & Wang (2017); Kim & Bengio (2017); Zhai et al. (2016); Haarnoja et al. (2017); Du & Mordatch (2019); Nijkamp et al. (2019)). Apart from DEM, there are also some other forms of deep explicit models based on restricted Boltzmann machines like deep belief networks (Hinton et al. (2006)) and deep Boltzmann machines (Salakhutdinov & Hinton (2009)).

The normalized constant under the energy function requires an intractable integral over all possible instances, which makes the model hard to learn via Maximum Likelihood Estimation (MLE). To solve this issue, some works propose to approximate the constant by MCMC methods (Geman & Geman (1984); Neal (2011)). However, MCMC requires an inner-loop samples in each training, which induces high computational costs. Another solution is to optimize an alternate surrogate loss function. For example, contrastive divergence (CD) (Liu & Wang (2017)) is proposed to measure how much KL divergence can be improved by running a small numbers of Markov chain steps towards the intractable likelihood, while score matching (SM) (Hyvärinen (2005)) detours the constant by minimizing the distance for gradients of log-likelihoods. Moreover, the intractable normalized constant makes it hard to sample from. To obtain an accurate samples from unnormalized densities, many studies propose to approximate the generation by diffusion-based processes, like generative flow (Nguyen et al. (2017)) and variational gradient descent (Liu & Wang (2016)). Also, a recent work (Hu et al. (2018)) leverages Stein discrepancy to design a neural sampler from unnormalized densities. The fundamental disadvantage of explicit model is that the energy-based learning is difficult to accurately capture the distribution of true samples due to the low manifold of real-world instances (Liu & Wang (2017)).

## A.2  IMPLICIT GENERATIVE MODELS

Implicit generative models focus on a generation mapping from random noises to generated samples. Such mapping function is often called as generator and possesses better flexibility compared with explicit models. Two typical implicit models are Variational Auto-Encoder (VAE) (Kingma & Welling (2014)) and Generative Adversarial Networks (GAN) (Goodfellow et al. (2014)). VAE introduces a latent variable and attempts to maximize the variational lower bound for likelihood of joint distribution of latent variable and observable variable, while GAN targets an adversarial game between the generator and a discriminator (or critic in WGAN) that aims at discriminating the generated and true samples. In this paper, we focus on GAN and its variants (e.g., WGAN (Arjovsky et al. (2017)), WGAN-GP (Gulrajani et al. (2017)), DCGAN (Radford et al. (2016)), etc.) as the implicit generative model and we leave the discussions on VAE as future work.

Two important issues concerning GAN and its variants are instability of training and local optima. The typical local optima for GAN can be divided into two categories: mode-collapse (the model fails to capture all the modes in data) and mode-redundance (the model generates modes that do not exist in data). Recently there are many attempts to solve these issues from various perspectives. One perspective is from regularization. Two typical regularization methods are likelihood-based and entropy-based regularization with the prominent examples Warde-Farley & Bengio (2017) and Li & Turner (2018) that respectively leverage denoising feature matching and implicit gradient approximation to enforce the regularization constraints. The likelihood and entropy regularizations could respectively help the generator to focus on data distribution and encourage more diverse samples, and a recent work (Tao et al. (2019)) uses Langevin dynamics to indicate that i) the entropy and likelihood regularizations are equivalent and share an opposite relationship in mathematics, and ii)

both regularizations would make the model converge to a surrogate point with a bias from original data distribution. Then Tao et al. (2019) proposes a variational annealing strategy to empirically unite two regularizations and tackle the biased distributions.

To deal with the instability issue, there are also some recent literatures from optimization perspectives and proposes different algorithms to address the non-convergence of minimax game optimization (for instance, Gemp & Mahadevan (2018); Liang & Stokes (2019); Gidel et al. (2019)). Moreover, the disadvantage of implicit models is the lack of explicit densities over instances, which disables the black-box generator to characterize the distributions behind data.

### A.3 ATTEMPTS TO COMBINE BOTH OF THE WORLDS

Recently, there are several studies that attempt to combine explicit and implicit generative models from different ways. For instance, Zhao et al. (2017) proposes energy-based GAN that leverages energy model as discriminator to distinguish the generated and true samples. The similar idea is also used by Kim & Bengio (2017) and Dai et al. (2017) which let the discriminator estimate a scaler energy value for each sample. Such discriminator is optimized to give high energy to generated samples and low energy to true samples while the generator aims at generating samples with low energy. The fundamental difference is that Zhao et al. (2017) and Dai et al. (2017) both aim at minimizing the discrepancy between distributions of generated and true samples while the motivation of Kim & Bengio (2017) is to minimize the KL divergence between estimated densities and true samples. Kim & Bengio (2017) adopts contrastive divergence (CD) to link MLE for energy model over true data with the adversarial training of energy-based GAN. However, both CD-based method and energy-based GAN have limited power for both generator and discriminator. Firstly, if the generated samples resemble true samples, then the gradients for discriminator given by true and generated samples are just the opposite and will counteract each other, and the training will stop before the discriminitor captures accurate data distribution. Second, since the objective boils down to minimizing the KL divergence (for Kim & Bengio (2017)) or Wasserstein distance (for Dai et al. (2017)) between model and true distributions, the issues concerning GAN (or WGAN) like training instability and mode-collapse would also bother these methods.

Another way for combination is by cooperative training. Xie et al. (2016a) (and its improved version Xie et al. (2018)) leverages the samples of generator as the MCMC initialization for energy-based model. The synthesized samples produced from finite-step MCMC are closer to the energy model and the generator is optimized to make the finite-step MCMC revise its initial samples. Also, a recent work Du et al. (2018) proposes to regard the explicit model as a teacher net who guides the training of implicit generator as a student net to produce samples that could overcome the mode-collapse issue. The main drawback of cooperative training is that they indirectly optimize the discrepancy between the generator and data distribution via the energy model as a 'mediator', which leads to a fact that once the energy model gets stuck in a local optimum (e.g., mode-collapse or mode-redundance) the training for the generator would be affected. In other words, the training for two models would constrain rather than exactly compensate each other. In Table 1, we do a high-level comparison among the above-mentioned generative models w.r.t the objectives. Different from other methods, our model considers three discrepancies simultaneously as a triangle to jointly train the generator and the estimator, enabling them to compensate and reinforce each other.

## B BACKGROUND FOR STEIN DISCREPANCY

Assume $q(\mathbf{x})$ to be a continuously differentiable density supported on $\mathcal{X} \subset \mathbb{R}^d$ and $\mathbf{f} : \mathbb{R}^d \to \mathbb{R}^{d'}$ a smooth vector function. Define $\mathcal{A}_q[\mathbf{f}(\mathbf{x})] = \nabla_{\mathbf{x}} \log q(\mathbf{x})\mathbf{f}(\mathbf{x})^\top + \nabla_{\mathbf{x}}\mathbf{f}(\mathbf{x})$ as a Stein operator. If $\mathbf{f}$ is a Stein class (satisfying some mild boundary conditions) then we have the following Stein identity property:

$$\mathbb{E}_{\mathbf{x}\sim q}[A_q[\mathbf{f}(\mathbf{x})]] = \mathbb{E}_{\mathbf{x}\sim q}[\nabla_{\mathbf{x}} \log q(\mathbf{x})\mathbf{f}(\mathbf{x})^\top + \nabla_{\mathbf{x}}\mathbf{f}(\mathbf{x})] = 0.$$

Such property induces the Stein discrepancy between distributions $\mathbb{P} : p(\mathbf{x})$ and $\mathbb{Q} : q(\mathbf{x})$, $\mathbf{x} \in \mathcal{X}$:

$$\mathcal{S}(\mathbb{Q}, \mathbb{P}) = \sup_{\mathbf{f}\in\mathcal{F}}\{\mathbb{E}_{\mathbf{x}\sim q}[A_p[\mathbf{f}(\mathbf{x})]] = \sup_{\mathbf{f}\in\mathcal{F}}\{\phi(\mathbb{E}_{\mathbf{x}\sim q}[\nabla_{\mathbf{x}} \log p(\mathbf{x})\mathbf{f}(\mathbf{x})^\top + \nabla_{\mathbf{x}}\mathbf{f}(\mathbf{x})])\}, \qquad (10)$$

where $\mathbf{f}$ is what we call *Stein critic* that exploits over function space $\mathcal{F}$ and if $\mathcal{F}$ is large enough then $\mathcal{S}(\mathbb{Q}, \mathbb{P}) = 0$ if and only if $\mathbb{Q} = \mathbb{P}$. Note that in (1), we do not need the normalized constant for $p(\mathbf{x})$ which enables Stein discrepancy to deal with unnormalized density.

If $\mathcal{F}$ is a unit ball in a Reproducing Kernel Hilbert Space (RKHS) with a positive definite kernel function $k(\cdot, \cdot)$, then the supremum in (1) would have a close form (see Liu et al. (2016); Chwialkowski et al. (2016); Oates et al. (2017) for more details):

$$\mathcal{S}_K(\mathbb{Q}, \mathbb{P}) = \mathbb{E}_{\mathbf{x}, \mathbf{x}' \sim q}[u_p(\mathbf{x}, \mathbf{x}')], \tag{11}$$

where $u_p(\mathbf{x}, \mathbf{x}') = \nabla_{\mathbf{x}} \log p(\mathbf{x})^\top k(\mathbf{x}, \mathbf{x}') \nabla_{\mathbf{x}} \log p(\mathbf{x}') + \nabla_{\mathbf{x}} \log p(\mathbf{x})^\top \nabla_{\mathbf{x}} k(x, \mathbf{x}') + \nabla_{\mathbf{x}} k(\mathbf{x}, \mathbf{x}')^\top \nabla_{\mathbf{x}} \log p(\mathbf{x}') + tr(\nabla_{\mathbf{x}, \mathbf{x}'} k(\mathbf{x}, \mathbf{x}'))$. The (11) gives the Kernel Stein Discrepancy (KSD).

## C   PROOFS OF RESULTS IN SECTION 3.2

### C.1   PROOF OF THEOREM 1

*Proof.* Using Kantorovich's duality, we rewrite the problem as

$$\min_{E, \mathbb{P}} \max_D \{ \mathbb{E}_{\mathbb{P}}[D] - \mathbb{E}_{\mathbb{P}_{\text{real}}}[D] + \lambda_1 \mathcal{S}(\mathbb{P}_{\text{real}}, \mathbb{P}_E) + \lambda_2 \mathcal{S}(\mathbb{P}, \mathbb{P}_E) \},$$

where the minimization with respect to $E$ is over all energy functions, the minimization with respect to $\mathbb{P}$ is over all probability distributions with continuous density, and the maximization is over all 1-Lipschitz continuous functions. From the definition of kernel Stein discrepancy

$$\mathcal{S}(\mathbb{P}, \mathbb{P}_E) = \mathbb{E}_{\mathbf{x}, \mathbf{x}' \in \mathbb{P}}[(\nabla_x \log \mathbb{P}(\mathbf{x}) - \nabla_x \log \mathbb{P}_E(\mathbf{x}))^\top k(\mathbf{x}, \mathbf{x}')(\nabla_x \log \mathbb{P}(\mathbf{x}') - \nabla_x \log \mathbb{P}_E(\mathbf{x}'))],$$

$\mathcal{S}(\mathbb{P}, \mathbb{P}_E)$ is infinite if $\mathbb{P}$ is not absolutely continuous with respect to $\mathbb{P}_E$. Hence, to minimize the objective, it suffices to consider those $\mathbb{P}$'s that are absolutely continuous with respect to $\mathbb{P}_E$. We set $h(\mathbf{x}) := d\mathbb{P}/d\mathbb{P}_E(\mathbf{x}) - 1$, where $d\mathbb{P}/d\mathbb{P}_E$ is the Radon-Nikodym derivative. It follows that the problem becomes

$$\min_{E, h} \max_D \left\{ \mathbb{E}_{\mathbb{P}_E}[D] + \mathbb{E}_{\mathbb{P}_E}[hD] - \mathbb{E}_{\mathbb{P}_{\text{real}}}[D] + \lambda_1 \mathcal{S}(\mathbb{P}_{\text{real}}, \mathbb{P}_E) \right.$$
$$\left. + \lambda_2 \cdot \mathbb{E}_{\mathbf{x}, \mathbf{x}' \sim \mathbb{P}}[\nabla_x \log(1 + h(\mathbf{x}))^\top k(\mathbf{x}, \mathbf{x}') \nabla_x \log(1 + h(\mathbf{x}'))] \right\},$$

where the minimization with respect to $h$ is over all $L^1(\mathbb{P}_E)$ functions with $\mathbb{P}_E$-expectation zero.

Fixing $E$, we claim that we can swap $\min_h$ and $\max_D$. Indeed, without loss of generality, we can restrict $D$ to be such that $D(\mathbf{x}_0) = 0$ for some element $\mathbf{x}_0$, as a constant shift does not change the value of $\mathbb{E}_{\mathbb{P}_E}[(1 + h)D] - \mathbb{E}_{\mathbb{P}_{\text{real}}}[D]$. The set of Lipschitz functions that vanish at $\mathbf{x}_0$ is a Banach space, and the subset of 1-Lipschitz functions is compact Weaver (1999). Moreover, $L^1(\mathbb{P}_E)$ is also a Banach space. The above verifies the condition of Sion's minimax theorem, and thus the claim is proved.

Swapping $\min_h$ and $\max_D$, we consider

$$\min_E \max_D \min_{h: \mathbb{E}_{\mathbb{P}_E}[h] = 0} \{ \mathbb{E}_{\mathbb{P}_E}[hD] + \lambda_2 \cdot \mathbb{E}_{\mathbf{x}, \mathbf{x}' \sim \mathbb{P}}[\nabla_x \log(1 + h(\mathbf{x}))^\top k(\mathbf{x}, \mathbf{x}') \nabla_x \log(1 + h(\mathbf{x}'))] \}$$

$$= \min_E \max_D \min_{h: \mathbb{E}_{\mathbb{P}_E}[h] = 0} \left\{ \mathbb{E}_{\mathbb{P}_E}[hD] + \lambda_2 \cdot \mathbb{E}_{\mathbf{x}, \mathbf{x}' \sim \mathbb{P}} \left[ \frac{\nabla_x h(\mathbf{x})^\top}{1 + h(\mathbf{x})} k(\mathbf{x}, \mathbf{x}') \frac{\nabla_x h(\mathbf{x}')}{1 + h(\mathbf{x}')} \right] \right\}$$

$$= \min_E \max_D \min_{h: \mathbb{E}_{\mathbb{P}_E}[h] = 0} \left\{ \mathbb{E}_{\mathbb{P}_E}[hD] + \lambda_2 \cdot \mathbb{E}_{\mathbf{x}, \mathbf{x}' \sim \mathbb{P}_E} \left[ \nabla_x h(\mathbf{x})^\top k(\mathbf{x}, \mathbf{x}') \nabla_x h(\mathbf{x}') \right] \right\},$$

where the second equality follows from the chain rule of the derivative. For the inner minimum, we have that

$$\min_{h: \mathbb{E}_{\mathbb{P}_E}[h] = 0} \left\{ \mathbb{E}_{\mathbb{P}_E}[hD] + \lambda_2 \cdot \mathbb{E}_{\mathbf{x}, \mathbf{x}' \sim \mathbb{P}_E} \left[ \nabla_x h(\mathbf{x})^\top k(\mathbf{x}, \mathbf{x}') \nabla_x h(\mathbf{x}') \right] \right\}$$

$$= \min_{r \geq 0} \min_{h: \mathbb{E}_{\mathbb{P}_E}[h] = 0} \left\{ \mathbb{E}_{\mathbb{P}_E}[hD] + \lambda_2 r^2 : \mathbb{E}_{\mathbf{x}, \mathbf{x}' \sim \mathbb{P}_E} \left[ \nabla_x h(\mathbf{x})^\top k(\mathbf{x}, \mathbf{x}') \nabla_x h(\mathbf{x}') \right] \leq r^2 \right\}$$

$$= \min_{r \geq 0} \min_{h: \mathbb{E}_{\mathbb{P}_E}[h] = 0} \left\{ r \mathbb{E}_{\mathbb{P}_E}[hD] + \lambda_2 r^2 : \mathbb{E}_{\mathbf{x}, \mathbf{x}' \sim \mathbb{P}_E} \left[ \nabla_x h(\mathbf{x})^\top k(\mathbf{x}, \mathbf{x}') \nabla_x h(\mathbf{x}') \right] \leq 1 \right\}$$

$$= \min_{r \geq 0} \left\{ \lambda_2 r^2 - r \|D\|_{H^{-1}(\mathbb{P}_E; k)} \right\}$$

$$= -\frac{1}{4\lambda_2} \|D\|_{H^{-1}(\mathbb{P}_E; k)},$$

where the first equality follows the introduction of the auxiliary variable $r$, the second equality follows a change of variable from $h$ to $rh$; and the second equality the third equality follows from the definition of the kernel Sobolev dual norm. Plugging in the previous equation yields the ideal result.

$\square$

## C.2 PROOF FOR THEOREM 2

*Proof.* Essentially, the result is a consequence of distributionally robust optimization with Wasserstein metric (Gao & Kleywegt, 2016; Gao et al., 2017). Here we provide a simplified version for completeness.

Using the definition of Stein discrepancy, we rewrite the problem as

$$\min_{E,\mathbb{P}} \max_{\mathbf{f}} \{\lambda_1 \mathcal{S}(\mathbb{P}_{\text{real}}, \mathbb{P}_E) + \lambda_2 \mathbb{E}_{\mathbf{y} \sim \mathbb{P}}[\mathcal{A}_{\mathbb{P}_E} \mathbf{f}(\mathbf{y})] + \mathcal{W}(\mathbb{P}_{\text{real}}, \mathbb{P})\}$$

Using the compactness of the Stein class and similar argument to the proof of Theorem 1, we can swap the minimization over $\mathbb{P}$ and the maximization over $\mathbf{f}$. Fixing $\mathbf{f}$, consider

$$\min_{\mathbb{P}} \{\lambda_2 \mathbb{E}_{\mathbf{y} \sim \mathbb{P}}[\mathcal{A}_{\mathbb{P}_E} \mathbf{f}(\mathbf{y})] + \mathcal{W}(\mathbb{P}_{\text{real}}, \mathbb{P})\}.$$

Using the definition of Wasserstein metric

$$\mathcal{W}(\mathbb{P}_{\text{real}}, \mathbb{P}) = \min_{\gamma} \mathbb{E}_{(\mathbf{x},\mathbf{y}) \sim \gamma}[\|\mathbf{x} - \mathbf{y}\|],$$

where the minimization is over all joint distributions of $(\mathbf{x}, \mathbf{y})$ with $\mathbf{x}$-marginal $\mathbb{P}_{\text{real}}$ and $\mathbf{y}$-marginal $\mathbb{P}$, we write the problem above as

$$\min_{\mathbb{P},\gamma} \{\mathbb{E}_{(\mathbf{x},\mathbf{y}) \sim \gamma} [\lambda_2 \mathcal{A}_{\mathbb{P}_E} \mathbf{f}(\mathbf{y}) + \|\mathbf{x} - \mathbf{y}\|]\},$$

where $\gamma$ has marginals $\mathbb{P}_{\text{real}}$ and $\mathbb{P}$. Since $\mathbb{P}$ is unconstrained, the above problem is further equivalent to

$$\min_{\gamma} \{\mathbb{E}_{(\mathbf{x},\mathbf{y}) \sim \gamma} [\lambda_2 \mathcal{A}_{\mathbb{P}_E} \mathbf{f}(\mathbf{y})] + \|\mathbf{x} - \mathbf{y}\|]\},$$

where the minimization is over all joint distributions of $(\mathbf{x}, \mathbf{y})$ with $\mathbf{x}$-marginal being $\mathbb{P}_{\text{real}}$. Using the law of total expectation, the problem above is equivalent to

$$\min_{\{\gamma_{\mathbf{x}}\}_{\mathbf{x}}} \mathbb{E}_{\mathbf{x} \sim \mathbb{P}_{\text{real}}} [\mathbb{E}_{\mathbf{y} \sim \gamma_{\mathbf{x}}} [\lambda_2 \mathcal{A}_{\mathbb{P}_E} \mathbf{f}(\mathbf{y}) + \|\mathbf{x} - \mathbf{y}\| \mid \mathbf{x}]]$$

$$= \mathbb{E}_{\mathbf{x} \sim \mathbb{P}_{\text{real}}} \left[ \min_{\gamma_{\mathbf{x}}} \{\mathbb{E}_{\mathbf{y} \sim \gamma_{\mathbf{x}}} [\lambda_2 \mathcal{A}_{\mathbb{P}_E} \mathbf{f}(\mathbf{y}) + \|\mathbf{x} - \mathbf{y}\| \mid \mathbf{x}]\} \right]$$

$$= \mathbb{E}_{\mathbf{x} \sim \mathbb{P}_{\text{real}}} \left[ \min_{\mathbf{y} \in \mathcal{X}} \{\lambda_2 \mathcal{A}_{\mathbb{P}_E} \mathbf{f}(\mathbf{y}) + \|\mathbf{x} - \mathbf{y}\|\} \right]$$

where the minimization in the first equation is over $\gamma_{\mathbf{x}}$, all conditional distributions of $\mathbf{y}$ given $\mathbf{x}$, and the exchanging of $\min$ and $\mathbb{E}$ follows from the interchangebability principle (Shapiro et al., 2009); the second equality holds because the infimum can be restricted to the set of point masses. We finally have the original problem is equivalent to

$$\min_{E} \max_{\mathbf{f}} \left\{ \lambda_1 \mathcal{S}(\mathbb{P}_{\text{real}}, \mathbb{P}_E) + \mathbb{E}_{\mathbf{x} \sim \mathbb{P}_{\text{real}}} \left[ \min_{\mathbf{y} \in \mathcal{X}} \{\lambda_2 \mathcal{A}_{\mathbb{P}_E} \mathbf{f}(\mathbf{y}) + \|\mathbf{x} - \mathbf{y}\|\} \right] \right\}.$$

Hence the proof is completed using the definition of Moreau-Yosida regularization. $\square$

## D PROOFS AND MORE DISCUSSIONS IN SECTION 4

### D.1 DETAILS FOR ONE-DIMENSIONAL CASE

For the analysis of 1-dim regularized WGAN in section 3.1.1, we assume a Gaussian likelihood function for sample $x$, $\mu(x) = \exp(-\frac{1}{2}(x - b)^2)$ which is up to a constant. Its parameter can be estimated by $b = \mathbb{E}[x]$. Then for generated sample $x' = \theta z$, we have $\mathbb{E}(\log \mu(\theta z)) = -\frac{1}{2}\mathbb{E}[z^2]\theta^2 +$

$\mathbb{E}[z]\mathbb{E}[x]\theta - \frac{1}{2}\mathbb{E}[x]^2$. Like the case in WGAN, we consider $\mathbb{E}[x] = \mathbb{E}[z] = 1$. Assume $\text{Var}[z] = 1$ and we have $\mathbb{E}[z^2] = 1 + \mathbb{E}[z]$. Hence, for the analysis on likelihood- (and entropy-) regularized WGAN, we can study the following system:

$$\min_\theta \max_\psi \psi - \psi \cdot \theta - \lambda(\theta^2 - \theta).$$

When $\lambda = 1$, the above objective degrades to (4); when $\lambda < 0$ (likelihood-regularization), the the gradient of regularization term pushes $\theta$ to shrink, which helps for convergence; when $\lambda > 0$ (entropy-regularization), the added term forms an amplifiying strength on $\theta$ and leads to divergence.

We proceed to consider 1-dim case of Stein Bridging with energy model $p_\phi(x) = \exp(-\frac{1}{2}x^2 - \phi x)$. If using KSD with kernel $k(x_1, x_2) = \mathbb{I}(x_1 = x_2)$, then $\mathcal{S}(\mathbb{P}_{real}, \mathbb{P}_E) = \mathbb{E}_{x_1, x_2}[(\nabla_{x_1} \log p_\phi(x_1) - \nabla_{x_1} \log \mu(x_1))k(x_1, x_2)(\nabla_{x_2} \log p_\phi(x_2) - \nabla_{x_2} \log \mu(x_2))] = \mathbb{E}_x[(\nabla_x \log p_\phi(x) - \nabla_x \log \mu(x))^2] = (\phi + \mathbb{E}[x])^2$. Similarly, one can obtain $\mathcal{S}(\mathbb{P}_G, \mathbb{P}_E) = (\phi + \theta\mathbb{E}[z])^2$. Therefore we arrive at the objective in (6)

$$\min_\theta \max_\psi \min_\phi \psi - \psi \cdot \theta + \frac{\lambda_1}{2}(1 + \phi)^2 + \frac{\lambda_2}{2}(\theta + \phi)^2. \tag{12}$$

Interestingly, the added terms $\frac{\lambda_1}{2}(1 + \phi)^2 + \frac{\lambda_2}{2}(\theta + \phi)^2$ in (6) and the original terms $\psi - \psi \cdot \theta$ in WGAN play both necessary roles to guarantee the convergence to the unique optimum points $[\psi^*, \theta^*, \phi^*] = [0, 1, -1]$. If we remove the critic and optimize $\theta$ and $\phi$ with the remaining loss terms, we would find that the training would converge but not necessarily to $[\psi^*, \theta^*] = [0, 1]$ (since the optimum points are not unique in this case). On the other hand, if we remove the estimator, the system degrades to (4) and would not converge to the unique optimum point $[\psi^*, \theta^*] = [0, 1]$. If we consider both of the world and optimize three terms together, the training would converge to a unique global optimum $[\psi^*, \theta^*, \phi^*] = [0, 1, -1]$.

### D.2 PROOF FOR PROPOSITION 1

*Proof.* Instead of directly studying the optimization for (6), we first prove the following problem will converge to the unique optimum,

$$\min_\theta \max_\psi \min_\phi \theta\psi + \theta\phi + \frac{1}{2}\theta^2 + \phi^2. \tag{13}$$

Applying alternate SGD we have the following iterations:

$$\psi_{t+1} = \psi_t + \eta * \theta_t,$$
$$\phi_{t+1} = \phi_t - \eta * (\theta_t + 2\phi_t) = (1 - 2\eta)\phi_t - \eta\theta_t,$$
$$\theta_{t+1} = \theta_t - \eta(\psi_{t+1} + \phi_{t+1} + \theta_t) = -\eta(1 - 2\eta)\phi_t + (1 - \eta)\theta_t - \eta\psi_t.$$

Then we obtain the relationship between adjacent iterations:

$$\begin{bmatrix} \psi_{t+1} \\ \phi_{t+1} \\ \theta_{t+1} \end{bmatrix} = \begin{bmatrix} 1 & 0 & \eta \\ 0 & 1 - 2\eta & -\eta \\ -\eta & -\eta(1 - 2\eta) & 1 - \eta \end{bmatrix} \cdot \begin{bmatrix} \psi_t \\ \phi_t \\ \theta_t \end{bmatrix} = M \cdot \begin{bmatrix} \psi_t \\ \phi_t \\ \theta_t \end{bmatrix}$$

We further calculate the eigenvalues for matrix $M$ and have the following equations (assume the eigenvalue as $\lambda$):

$$(\lambda - 1)^3 + 3\eta(\lambda - 1)^2 + 2\eta^2(1 + \eta)(\lambda - 1) + 2\eta^3 = 0.$$

One can verify that the solutions to the above equation satisfy $|\lambda| < \sqrt{(1 - \eta + \eta^2)(1 + \eta - \eta^2)}$.

Then we have the following relationship

$$\left\| \begin{bmatrix} \psi_{t+1} \\ \phi_{t+1} \\ \theta_{t+1} \end{bmatrix} \right\|_2^2 = \left\| [\psi_t \quad \phi_t \quad \theta_t] \cdot M^\top M \cdot \begin{bmatrix} \psi_t \\ \phi_t \\ \theta_t \end{bmatrix} \right\|_2^2 \leq \lambda_m^2 \cdot \left\| \begin{bmatrix} \psi_t \\ \phi_t \\ \theta_t \end{bmatrix} \right\|_2^2$$

where $\lambda_m$ denotes the eigenvalue with the maximum absolute value of matrix $M$. Hence, we have

$$\psi_{t+1}^2 + \phi_{t+1}^2 + \theta_{t+1}^2 \leq (1 - \eta + \eta^2)(1 + \eta - \eta^2)[\psi_t^2 + \phi_t^2 + \theta_t^2].$$

We proceed to replace $\psi$, $\phi$ and $\theta$ in (13) by $\psi'$, $\phi'$ and $\theta'$ respectively and conduct a change of variable: let $\theta' = 1 - \theta$ and $\phi' = -1 - \phi$. Then we get the conclusion in the proposition.

$\square$

### D.3 GENERALIZATION TO BILINEAR SYSTEMS

Our analysis in the one-dimension case inspires us that we can add affiliated variable to modify the objective and stabilize the training for general bilinear system. The bilinear system is of wide interest for researchers focusing on stability of GAN training (Goodfellow (2017); Liang & Stokes (2019); Gidel et al. (2019); Gemp & Mahadevan (2018)). The general bilinear function can be written as

$$F(\psi, \theta) = \theta^\top \mathbf{A} \psi - \mathbf{b}^\top \theta - \mathbf{c}^\top \psi, \tag{14}$$

where $\psi, \theta$ are both $r$-dimensional vectors and the objective is $\min_{\theta} \max_{\psi} F(\psi, \theta)$ which can be seen as a basic form of various GAN objectives. Unfortunately, if we directly use simultaneous (resp. alternate) SGD to optimize such objectives, one can obtain divergence (resp. fluctuation). To solve the issue, some recent papers propose several optimization algorithms, like extrapolation from the past (Gidel et al. (2019)), crossing the curl (Gemp & Mahadevan (2018)) and consensus optimization (Liang & Stokes (2019)). Also, Liang & Stokes (2019) shows that it is the interaction term which generates non-zero values for $\nabla_{\theta\psi} F$ and $\nabla_{\psi\theta} F$ that leads to such instability of training. Different from previous works that focused on algorithmic perspective, we propose to add new affiliated variables which modify the objective function and allow the SGD algorithm to achieve convergence without changing the optimum points.

Based on the minimax objective of (14) we add affiliated $r$-dimensional variable $\phi$ (corresponding to the estimator in our model) the original system and tackle the following problem:

$$\min_{\theta} \max_{\psi} \min_{\phi} F(\psi, \theta) + \alpha H(\phi, \theta), \tag{15}$$

where $H(\phi, \theta) = \frac{1}{2}(\theta + \phi)^\top \mathbf{B}(\theta + \phi)$, $\mathbf{B} = (\mathbf{A}\mathbf{A}^\top)^{\frac{1}{2}}$ and $\alpha$ is a non-negative constant. Theoretically, the new problem keeps the optimum points of (14) unchanged. Let $L(\psi, \phi, \theta) = F(\psi, \theta) + \alpha G(\phi, \theta)$

**Proposition 2.** *Assume the optimum point of $\min_{\theta} \max_{\psi} F(\psi, \theta)$ are $[\psi^*, \theta^*]$, then the optimum points of (15) would be $[\psi^*, \theta^*, \phi^*]$ where $\phi^* = -\theta^*$.*

*Proof.* The condition tells us that $\nabla_{\theta} F(\psi^*, \theta) = 0$ and $\nabla_{\psi} F(\psi, \theta^*) = 0$. Then we derive the gradients for $L(\psi, \phi, \theta)$,

$$\nabla_{\psi} L(\psi^*, \phi, \theta) = \nabla_{\theta} F(\psi^*, \theta) = 0, \tag{16}$$

$$\nabla_{\theta} L(\psi, \phi, \theta^*) = \nabla_{\theta} F(\psi, \theta^*) + \nabla_{\theta} H(\phi, \theta^*) = \frac{1}{2}(\mathbf{B} + \mathbf{B}^\top)(\theta^* + \phi), \tag{17}$$

$$\nabla_{\phi} L(\psi, \phi, \theta) = \nabla_{\phi} H(\phi, \theta) = \frac{1}{2}(\mathbf{B} + \mathbf{B}^\top)(\phi + \theta), \tag{18}$$

Combining (17) and (18) we get $\phi^* = -\theta^*$. Hence, the optimum point of (15) is $[\psi^*, \theta^*, \phi^*]$ where $\phi^* = -\theta^*$. $\square$

The advantage of the new problem is that it can be solved by SGD algorithm and guarantees convergence theoretically. We formulate the results in the following theorem.

**Theorem 4.** *For problem $\min_{\theta} \max_{\psi} \min_{\phi} L(\psi, \phi, \theta)$ using alternate SGD algorithm, i.e.,*

$$\begin{aligned}
\psi_{t+1} &= \psi_t + \eta \nabla_{\psi} L(\theta_t, \psi_t, \phi_t), \\
\phi_{t+1} &= \phi_t - \eta \nabla_{\phi} L(\theta_t, \psi_{t+1}, \phi_t), \\
\theta_{t+1} &= \theta_t - \eta \nabla_{\theta} L(\theta_t, \psi_{t+1}, \phi_{t+1}),
\end{aligned} \tag{19}$$

*we can achieve convergence to $[\psi^*, \theta^*, \phi^*]$ where $\phi^* = -\theta^*$ with at least linear rate of $(1 - \eta_1 + \eta_2^2)(1 + \eta_2 - \eta_1^2)$ where $\eta_1 = \eta\sigma_{min}$, $\eta_2 = \eta\sigma_{max}$ and $\sigma_{min}$ (resp. $\sigma_{max}$) denotes the maximum (resp. minimum) singular value of matrix $\mathbf{A}$.*

To prove Theorem 3, we can prove a more general argument.

**Lemma 1.** *If we consider any first-order optimization method on (15), i.e.,*

$$\psi_{t+1} \in \psi_0 + span(L(\psi_0, \phi, \theta), \cdots, F(\psi_t, \phi, \theta)), \forall t \in \mathbb{N},$$
$$\phi_{t+1} \in \psi_0 + span(L(\psi, \phi_0, \theta), \cdots, L(\psi, \phi_t, \theta)), \forall t \in \mathbb{N},$$
$$\theta_{t+1} \in \psi_0 + span(L(\psi, \phi, \theta_0), \cdots, L(\psi, \phi, \theta_t)), \forall t \in \mathbb{N},$$

*Then we have*

$$\widetilde{\psi}_t = \mathbf{V}^\top(\psi_t - \psi^*), \quad \widetilde{\phi}_t = \mathbf{U}^\top(\phi_t - \phi^*), \quad \widetilde{\theta}_t = \mathbf{U}^\top(\theta_t - \theta^*),$$

*where $\mathbf{U}$ and $\mathbf{V}$ are the singular vectors decomposed by matrix $\mathbf{A}$ using SVD decomposition, i.e., $\mathbf{A} = \mathbf{UDV}^\top$ and the triple $([\widetilde{\psi}_t]_i, [\widetilde{\phi}_t]_i, [\widetilde{\theta}_t]_i)_{1 \leq i \leq r}$ follows the update rule with step size $\sigma_i \eta$ as the same optimization method on a unidimensional problem*

$$\min_\theta \max_\psi \min_\phi \theta\psi + \theta\phi + \frac{1}{2}\theta^2 + \frac{1}{2}\phi^2, \tag{20}$$

*with step size $\eta$, where $\sigma_i$ denotes the $i$-th singular value on the diagonal of $\mathbf{D}$.*

*Proof.* The proof is extended from the proof of Lemma 3 in Gidel et al. (2019). The general class of first-order optimization methods derive the following updations:

$$\psi_{t+1} = \psi_0 + \sum_{s=0}^{t+1} \rho_{st}(\mathbf{A}^\top \theta_s - \mathbf{c}) = \psi_0 + \sum_{s=0}^{t+1} \rho_{st}\mathbf{A}^\top(\theta_s - \theta^*),$$

$$\phi_{t+1} = \phi_0 + \frac{1}{2}\sum_{s=0}^{t+1} \delta_{st}(\mathbf{B} + \mathbf{B}^\top)(\theta_s + \phi_s),$$

$$\theta_{t+1} = \theta_0 + \sum_{s=0}^{t+1} \mu_{st}[\mathbf{A}(\psi_s - \psi^*) + \frac{1}{2}(\mathbf{B} + \mathbf{B}^\top)(\theta_s + \phi_s)],$$

where $\rho_{st}, \delta_{st}, \mu_{st} \in \mathbb{R}$ depend on specific optimization method (for example, in SGD, $\rho_{tt} = \delta_{tt} = \mu_{tt}$ remain as a non-zero constant for $\forall t$ and other coefficients are zero).

Using SVD $\mathbf{A} = \mathbf{UDV}^\top$ and the fact $\theta^* = -\phi^*$, $\mathbf{B} = (\mathbf{UDD}^\top\mathbf{U}^\top) = \mathbf{D}$, we have

$$\mathbf{V}^\top(\psi_{t+1} - \psi^*) = \mathbf{V}^\top(\psi_0 - \psi^*) + \sum_{s=0}^{t+1} \rho_{st}\mathbf{D}^\top\mathbf{U}^\top(\theta_s - \theta^*)$$

$$\mathbf{U}^\top(\phi_{t+1} - \phi^*) = \mathbf{U}^\top(\phi_0 - \phi^*) + \sum_{s=0}^{t+1} \delta_{st}\mathbf{U}^\top\mathbf{D}(\theta_s - \theta^*) + \mathbf{U}^\top\mathbf{D}(\phi_s - \phi^*),$$

$$\mathbf{U}^\top(\theta_{t+1} - \theta^*) = \mathbf{U}^\top(\theta_0 - \theta^*) + \sum_{s=0}^{t+1} \rho_{st}[\mathbf{DV}^\top(\psi_s - \psi^*) + \mathbf{U}^\top\mathbf{D}(\theta_s - \theta^*) + \mathbf{U}^\top\mathbf{D}(\phi_s - \phi^*)],$$

and equivalently,

$$\widetilde{\psi}_{t+1} = \widetilde{\psi}_0 + \sum_{s=0}^{t+1} \rho_{st}\mathbf{D}^\top\widetilde{\theta}_t, \quad \widetilde{\phi}_t = \widetilde{\phi}_0 + \sum_{s=0}^{t+1} \delta_{st}\mathbf{D}(\widetilde{\theta}_t + \widetilde{\phi}_t),$$

$$\widetilde{\theta}_{t+1} = \widetilde{\theta}_0 + \sum_{s=0}^{t+1} \rho_{st}\mathbf{D}(\widetilde{\psi}_t + \widetilde{\theta}_t + \widetilde{\phi}_t).$$

Note that $\mathbf{D}$ is a rectangular matrix with non-zero elements on a diagonal block of size $r$. Hence, the above $r$-dimensional problem can be reduced to $r$ unidimensional problems:

$$[\widetilde{\psi}_{t+1}]_i = [\widetilde{\psi}_0]_i + \sum_{s=0}^{t+1} \rho_{st}\sigma_i[\widetilde{\theta}_t]_i, \quad [\widetilde{\phi}_t]_i = [\widetilde{\phi}_0]_i + \sum_{s=0}^{t+1} \delta_{st}\sigma_i([\widetilde{\theta}_t]_i + [\widetilde{\phi}_t]_i),$$

$$[\widetilde{\theta}_{t+1}]_i = [\widetilde{\theta}_0]_i + \sum_{s=0}^{t+1} \rho_{st}\sigma_i([\widetilde{\psi}_t]_i + [\widetilde{\theta}_t]_i + [\widetilde{\phi}_t]_i).$$

The above iterations can be conducted independently in each dimension where the optimization in $i$-th dimension follows the same updating rule with step size $\sigma_i \eta$ as problem in (20). □

Furthermore, since problem (20) can achieve convergence with a linear rate of $(1-\eta+\eta^2)(1+\eta-\eta^2)$ using alternate SGD (the proof is similar to that of ((13))), the multi-dimensional problem in (15) can achieve convergence by SGD with at least a rate of $(1-\eta_1+\eta_2^2)(1+\eta_2-\eta_1^2)$ where $\eta_1 = \eta\sigma_{max}$, $\eta_2 = \eta\sigma_{min}$ and $\sigma_{max}$ (resp. $\sigma_{min}$) denotes the maximum (resp. minimum) singular value of matrix $\mathbf{A}$. We conclude the proof for Theorem 4.

Theorem 4 suggests that the added term $H(\boldsymbol{\phi}, \boldsymbol{\theta})$ with affiliated variables $\phi$ could help the SGD algorithm achieve convergence to the the same optimum points as directly optimizing $F(\boldsymbol{\psi}, \boldsymbol{\theta})$. Our method is related to consensus optimization algorithm (Liang & Stokes (2019)) which adds a regularization term $\|\nabla_{\boldsymbol{\theta}}F(\boldsymbol{\psi}, \boldsymbol{\theta})\| + \|\nabla_{\boldsymbol{\psi}}F(\boldsymbol{\psi}, \boldsymbol{\theta})\|$ to (14) resulting extra quadratic terms for $\boldsymbol{\theta}$ and $\boldsymbol{\psi}$. The disadvantage of such method is the requirement of Hessian matrix of $F(\boldsymbol{\psi}, \boldsymbol{\theta})$ which is computational expensive for high-dimensional data. By contrast, our solution only requires the first-order derivatives.

### D.4 STRONGLY CONVEXITY

In section 3.1.2, we assume $H(\boldsymbol{\theta}, \boldsymbol{\phi})$ as a $\mu$-strongly convex function which indicates that it satisfies the conditions:

$$(\nabla_{\boldsymbol{\theta}} H(\boldsymbol{\theta}, \cdot) - \nabla_{\boldsymbol{\theta}} H(\boldsymbol{\theta}', \cdot))^\top (\boldsymbol{\theta} - \boldsymbol{\theta}') \geq \mu\|\boldsymbol{\theta} - \boldsymbol{\theta}'\|_2^2, \forall \boldsymbol{\theta}, \boldsymbol{\theta}' \in \Omega_{\boldsymbol{\theta}},$$

$$(\nabla_{\boldsymbol{\phi}} H(\cdot, \boldsymbol{\phi}) - \nabla_{\boldsymbol{\phi}} H(\cdot, \boldsymbol{\phi}'))^\top (\boldsymbol{\phi} - \boldsymbol{\phi}') \geq \mu\|\boldsymbol{\phi} - \boldsymbol{\phi}'\|_2^2, \forall \boldsymbol{\phi}, \boldsymbol{\phi}' \in \Omega_{\boldsymbol{\phi}}.$$

Bedises, $F(\boldsymbol{\theta}, \boldsymbol{\psi})$ is $\mu$-strongly convex for $\boldsymbol{\theta}$ and $\mu$-strongly concave for $\psi$ so it satisfies:

$$(\nabla_{\boldsymbol{\theta}} F(\boldsymbol{\theta}, \cdot) - \nabla_{\boldsymbol{\theta}} F(\boldsymbol{\theta}', \cdot))^\top (\boldsymbol{\theta} - \boldsymbol{\theta}') \geq \mu\|\boldsymbol{\theta} - \boldsymbol{\theta}'\|_2^2, \forall \boldsymbol{\theta}, \boldsymbol{\theta}' \in \Omega_{\boldsymbol{\theta}},$$

$$(\nabla_{\boldsymbol{\psi}} F(\cdot, \boldsymbol{\psi}') - \nabla_{\boldsymbol{\psi}} F(\cdot, \boldsymbol{\psi}))^\top (\boldsymbol{\psi} - \boldsymbol{\psi}') \geq \mu\|\boldsymbol{\psi} - \boldsymbol{\psi}'\|_2^2, \forall \boldsymbol{\psi}, \boldsymbol{\psi}' \in \Omega_{\boldsymbol{\psi}}.$$

In section 3.1.2, we also define $h(\boldsymbol{\omega}_h) = \nabla_{\boldsymbol{\theta}} H + \nabla_{\boldsymbol{\phi}} H$ and $f(\boldsymbol{\omega}_f) = \nabla_{\boldsymbol{\theta}} F - \nabla_{\boldsymbol{\psi}} F$, so the above condition can be written in a more compact form,

$$(h(\boldsymbol{\omega}_h) - h(\boldsymbol{\omega}_h'))^\top (\boldsymbol{\omega}_h - \boldsymbol{\omega}_h') \geq \mu\|\boldsymbol{\omega}_h - \boldsymbol{\omega}_h'\|_2^2, \forall \boldsymbol{\omega}_h, \boldsymbol{\omega}_h' \in \Omega_h,$$

$$(f(\boldsymbol{\omega}_f) - f(\boldsymbol{\omega}_f'))^\top (\boldsymbol{\omega}_f - \boldsymbol{\omega}_f') \geq \mu\|\boldsymbol{\omega}_f - \boldsymbol{\omega}_f'\|_2^2, \forall \boldsymbol{\omega}_f, \boldsymbol{\omega}_f' \in \Omega_f.$$

### D.5 PROOF FOR THEOREM 2

The proof relies on two lemmas,

**Lemma 2.** *For any $\boldsymbol{\omega} \in \Omega$ and $\boldsymbol{\omega}^+ = P_\Omega(\boldsymbol{\omega} + \boldsymbol{u})$, then we have*

$$\|\boldsymbol{\omega}^+ - \boldsymbol{\omega}\|_2^2 \leq u^\top(\boldsymbol{\omega}^+ - \boldsymbol{\omega}).$$

*Proof.* Since $\boldsymbol{\omega}^+$ is a projection of $\boldsymbol{\omega} + \boldsymbol{u}$ on a convex set $\Omega$, we have

$$(\boldsymbol{\omega}^+ - (\boldsymbol{\omega} + \boldsymbol{u}))^\top (\boldsymbol{\omega}^+ - \boldsymbol{\omega}) \leq 0. \tag{21}$$

Rearranging the above inequality one can easily get the lemma. □

**Lemma 3.** *If function $\Phi(\boldsymbol{\omega})$ is $\mu$-strongly convex, we have*

$$\mu\|\boldsymbol{\omega} - \boldsymbol{\omega}^*\|_2^2 \leq \nabla\Phi(\boldsymbol{\omega})^\top (\boldsymbol{\omega} - \boldsymbol{\omega}^*).$$

*Similarly, if $\Phi(\boldsymbol{\omega})$ is $\mu$-strongly concave, we have $\mu(\boldsymbol{\omega} - \boldsymbol{\omega}^*) \leq \nabla - \Phi(\boldsymbol{\omega})^\top (\boldsymbol{\omega} - \boldsymbol{\omega}^*)$.*

*Proof.* By optimality of $\boldsymbol{\omega}^*$, we have

$$\nabla\Phi(\boldsymbol{\omega}^*)^\top (\boldsymbol{\omega} - \boldsymbol{\omega}^*) \geq 0.$$

Since $\Phi$ is $\mu$-convex, we can further derive

$$\mu\|\boldsymbol{\omega} - \boldsymbol{\omega}^*\|_2^2 \leq \nabla\Phi(\boldsymbol{\omega}^*)^\top (\boldsymbol{\omega} - \boldsymbol{\omega}^*) + \mu\|\boldsymbol{\omega} - \boldsymbol{\omega}^*\|_2^2 \leq \nabla\Phi(\boldsymbol{\omega})^\top (\boldsymbol{\omega} - \boldsymbol{\omega}^*).$$

□

*Proof.* (Proof for Theorem 3) We apply Lemma 2 to (9) with $(\boldsymbol{\omega}, \boldsymbol{u}, \boldsymbol{\omega}^+) = (\boldsymbol{\omega}_f^{t+1/2}, -\eta f(\boldsymbol{\omega}_f^{t+1/2}), \boldsymbol{\omega}_f^{t+1})$ and we have

$$\|\boldsymbol{\omega}_f^{t+1} - \boldsymbol{\omega}_f^{t+1/2}\| \leq -\eta f(\boldsymbol{\omega}_f^{t+1/2})^\top (\boldsymbol{\omega}_f^{t+1} - \boldsymbol{\omega}_f^{t+1/2}) \tag{22}$$

Then we have

$$\begin{aligned}
\|\boldsymbol{\omega}_f^{t+1} - \boldsymbol{\omega}_f^*\|_2^2 &= \|\boldsymbol{\omega}_f^{t+1/2} - \boldsymbol{\omega}_f^* + \boldsymbol{\omega}_f^{t+1} - \boldsymbol{\omega}_f^{t+1/2}\|_2^2 \\
&\leq 2\|\boldsymbol{\omega}_f^{t+1/2} - \boldsymbol{\omega}_f^*\|_2^2 + 2\|\boldsymbol{\omega}_f^{t+1} - \boldsymbol{\omega}_f^{t+1/2}\|_2^2 \quad (by \quad \|a+b\|_2^2 \leq 2\|a\|_2^2 + 2\|b\|_2^2) \\
&\leq 2\|\boldsymbol{\omega}_f^{t+1/2} - \boldsymbol{\omega}_f^*\|_2^2 - 2\eta f(\boldsymbol{\omega}_f^{t+1/2})^\top (\boldsymbol{\omega}_f^{t+1} - \boldsymbol{\omega}_f^{t+1/2}).
\end{aligned} \tag{23}$$

According to Lemma 3, we have

$$2\eta f(\boldsymbol{\omega}_f^{t+1/2})^\top (\boldsymbol{\omega}_f^{t+1/2} - \boldsymbol{\omega}_f^*) \geq 2\eta\mu \|\boldsymbol{\omega}_f^{t+1/2} - \boldsymbol{\omega}_f^*\|_2^2. \tag{24}$$

Plug (24) into (23) and we get

$$\|\boldsymbol{\omega}_f^{t+1} - \boldsymbol{\omega}_f^*\|_2^2 \leq (2 - 2\eta\mu)\|\boldsymbol{\omega}_f^{t+1/2} - \boldsymbol{\omega}_f^*\|_2^2. \tag{25}$$

The above inequality is equivalent to

$$\|\boldsymbol{\theta}^{t+1} - \boldsymbol{\theta}^*\|_2^2 + \|\boldsymbol{\psi}^{t+1} - \boldsymbol{\psi}^*\|_2^2 \leq (2 - 2\eta\mu)(\|\boldsymbol{\theta}^{t+1/2} - \boldsymbol{\theta}^*\|_2^2 + \|\boldsymbol{\psi}^t - \boldsymbol{\psi}^*\|_2^2). \tag{26}$$

Similarly, one can obtain

$$\|\boldsymbol{\omega}_h^{t+1} - \boldsymbol{\omega}_h^*\|_2^2 \leq (2 - 2\eta\mu)\|\boldsymbol{\omega}_h^{t+1/2} - \boldsymbol{\omega}_h^*\|_2^2, \tag{27}$$

i.e.,

$$\|\boldsymbol{\theta}^{t+1/2} - \boldsymbol{\theta}^*\|_2^2 + \|\boldsymbol{\phi}^{t+1} - \boldsymbol{\phi}^*\|_2^2 \leq (2 - 2\eta\mu)(\|\boldsymbol{\theta}^t - \boldsymbol{\theta}^*\|_2^2 + \|\boldsymbol{\phi}^t - \boldsymbol{\phi}^*\|_2^2). \tag{28}$$

Combining (26) and (28) we have

$$\begin{aligned}
\|\boldsymbol{\omega}^{t+1} - \boldsymbol{\omega}^*\|_2^2 &= \|\boldsymbol{\theta}^{t+1} - \boldsymbol{\theta}^*\|_2^2 + \|\boldsymbol{\psi}^{t+1} - \boldsymbol{\psi}^*\|_2^2 + \|\boldsymbol{\phi}^{t+1} - \boldsymbol{\phi}^*\|_2^2 \\
&\leq (1 - 2\eta\mu)\|\boldsymbol{\theta}^t - \boldsymbol{\theta}^*\|_2^2 + (2 - 2\eta\mu)\|\boldsymbol{\psi}^{t+1} - \boldsymbol{\psi}^*\|_2^2 + (2 - 2\eta\mu)\|\boldsymbol{\phi}^{t+1} - \boldsymbol{\phi}^*\|_2^2 \\
&\leq (2 - 2\eta\mu)\|\boldsymbol{\omega}^t - \boldsymbol{\omega}^*\|_2^2.
\end{aligned} \tag{29}$$

Hence, if $\frac{1}{2\mu} < \eta < \frac{1}{\mu}$ we have $0 < 2 - 2\eta\mu < 1$ and $\|\boldsymbol{\omega}^t - \boldsymbol{\omega}^*\|_2^2 \leq (2 - 2\eta\mu)^t \|\boldsymbol{\omega}^0 - \boldsymbol{\omega}^*\|_2^2$  □

# E   DETAILS FOR EXPERIMENT SETUP

## E.1   SYNTHETIC DATASETS

We provide the details for two synthetic datasets. The Two-Circle dataset consists of 24 Gaussian mixtures where 8 of them are located in an inner circle with radius $r_1 = 4$ and 16 of them lie in an outer circle with radius $r_2 = 8$. For each Gaussian component, the covariance matrix is $\begin{pmatrix} 0.2 & 0 \\ 0 & 0.2 \end{pmatrix} = \sigma_1 \mathbf{I}$ and the mean value is $[r_1 \cos t, r_1 \sin t]$, where $t = \frac{2\pi \cdot k}{8}$, $k = 1, \cdots, 8$, for the inner circle, and $[r_2 \cos t, r_2 \sin t]$, where $t = \frac{2\pi \cdot k}{16}$, $k = 1, \cdots, 16$ for the outer circle. We sample $N_1 = 2000$ points as true observed samples for model training. In section 5.5, we consider noised data scenario. In this case, we randomly add $n$ noise points sampled from Gaussian distribution $\mathcal{N}(\mathbf{0}, \sigma_0 \mathbf{I})$ where $\sigma_0 = 2$ to the original true samples. Here we set $n = [40, 100, 160, 300, 400, 600, 800, 1000]$.

The Two-Spiral dataset contains 100 Gaussian mixtures whose centers locate on two spiral-shaped curves. For each Gaussian component, the covariance matrix is $\begin{pmatrix} 0.5 & 0 \\ 0 & 0.5 \end{pmatrix} = \sigma_2 \mathbf{I}$ and the mean value is $[-c_1 \cos c_1, c_1 \sin c_1]$, where $c_1 = \frac{2\pi}{3} + linspace(0, 0.5, 50) \cdot 2\pi$, for one spiral, and $[c_2 \cos c_2, -c_2 \sin c_2]$, where $c_2 = \frac{2\pi}{3} + linspace(0, 0.5, 50) \cdot 2\pi$ for another spiral. We sample $N_2 = 5000$ points as true observed samples. In section 5.5, we consider insufficient data scenario. In this case, the sample size $N_2$ is reduced to $[100, 200, 300, 500, 700, 1000, 2000]$.

| $\mathcal{D}_1$ | $\mathcal{D}_2$ | $\mathcal{D}_3$ | Objective |
|---|---|---|---|
| $\mathcal{W}$ | $\mathcal{S}$ | $\mathcal{S}$ | $\min_\theta \min_\phi \max_\psi \max_\pi \mathbb{E}_{\mathbf{x}\sim\mathbb{P}_{data}}[d_\psi(\mathbf{x})] - \mathbb{E}_{\mathbf{z}\sim p_0}[d_\psi(G_\theta(\mathbf{z}))]$ $+\lambda_1 \mathbb{E}_{\mathbf{x}\sim\mathbb{P}_{data}}[\mathcal{A}_{p_\phi}[\mathbf{f}_\pi(\mathbf{x})]] + \lambda_2 \mathbb{E}_{\mathbf{z}\sim\mathfrak{l}_0}[\mathcal{A}_{p_\phi}[\mathbf{f}_\pi(G_\theta(\mathbf{z}))]]$ |
| $\mathcal{W}$ | $\mathcal{S}_k$ | $\mathcal{S}_k$ | $\min_\theta \min_\phi \max_\psi \mathbb{E}_{\mathbf{x}\sim\mathbb{P}_{data}}[d_\psi(\mathbf{x})] - \mathbb{E}_{\mathbf{z}\sim p_0}[d_\psi(G_\theta(\mathbf{z}))]$ $+\lambda_1 \mathbb{E}_{\mathbf{x},\mathbf{x}'\sim\mathbb{P}_{data}}[u_{p_\phi}(x,x')] + \lambda_2 \mathbb{E}_{\mathbf{z},\mathbf{z}'\sim p_0}[u_{p_\phi}(G_\theta(\mathbf{z}),G_\theta(\mathbf{z}'))]$ |
| $\mathcal{JS}$ | $\mathcal{S}$ | $\mathcal{S}$ | $\min_\theta \min_\phi \max_\psi \max_\pi \mathbb{E}_{\mathbf{x}\sim\mathbb{P}_r}[\log(d_\psi(\mathbf{x}))] + \mathbb{E}_{\mathbf{z}\sim p_0}[\log(1 - d_\psi(G_\theta(\mathbf{z})))]$ $+\lambda_1 \mathbb{E}_{\mathbf{x}\sim\mathbb{P}_{data}}[\mathcal{A}_{p_\phi}[\mathbf{f}_\pi(\mathbf{x})]] + \lambda_2 \mathbb{E}_{\mathbf{z}\sim\mathfrak{l}_0}[\mathcal{A}_{p_\phi}[\mathbf{f}_\pi(G_\theta(\mathbf{z}))]]$ |
| $\mathcal{JS}$ | $\mathcal{S}_k$ | $\mathcal{S}_k$ | $\min_\theta \min_\phi \max_\psi \mathbb{E}_{\mathbf{x}\sim\mathbb{P}_r}[\log(d_\psi(\mathbf{x}))] + \mathbb{E}_{\mathbf{z}\sim p_0}[\log(1 - d_\psi(G_\theta(\mathbf{z})))]$ $+\lambda_1 \mathbb{E}_{\mathbf{x},\mathbf{x}'\sim\mathbb{P}_{data}}[u_{p_\phi}(x,x')] + \lambda_2 \mathbb{E}_{\mathbf{z},\mathbf{z}'\sim p_0}[u_{p_\phi}(G_\theta(\mathbf{z}),G_\theta(\mathbf{z}'))]$ |

Table 4: Objectives for different specifications of $\mathcal{D}_1(\mathbb{P}_{\text{real}}, \mathbb{P}_G)$, $\mathcal{D}_2(\mathbb{P}_{\text{real}}, \mathbb{P}_E)$ and $\mathcal{D}_3(\mathbb{P}_G, \mathbb{P}_E)$. We specify $\mathcal{D}_1$ as Wasserstein distance or JS divergence in our paper and for $\mathcal{D}_2$ and $\mathcal{D}_3$ we consider the general Stein discrepancy or kernel Stein discrepancy. Here we use $\mathcal{W}$, $\mathcal{JS}$ to denote Wasserstein distance and JS divergence respectively, and $\mathcal{S}$, $\mathcal{S}_k$ to represent general Stein discrepancy and kernel Stein discrepancy respectively. We omit the gradient penalty term for Wasserstein distance here but use it in experiments.

### E.2 MODEL SPECIFICATIONS AND TRAINING ALGORITHM

In different tasks, we consider different model specifications in order to meet the demand of capacify as well as test the effectiveness under various settings. Our proposed framework (3) adopts Wasserstein distance for the first term and two Stein discrepancies for the second and the third terms. We can write (3) as a more general form

$$\min_{\theta,\phi} \mathcal{D}_1(\mathbb{P}_{\text{real}}, \mathbb{P}_G) + \lambda_1 \mathcal{D}_2(\mathbb{P}_{\text{real}}, \mathbb{P}_E) + \lambda_2 \mathcal{D}_3(\mathbb{P}_G, \mathbb{P}_E), \tag{30}$$

where $\mathcal{D}_1$, $\mathcal{D}_2$, $\mathcal{D}_3$ denote three general discrepancy measures for distributions. As stated in our remark, $\mathcal{D}_1$ can be specified as arbitrary discrepancy measures for implicit generative models. Here we also use JS divergence, the objective for valina GAN. To well distinguish them, we call the model using Wasserstein distance (resp. JS divergence) as Joint-W (resp. Joint-JS) in our experiments. On the other hand, the two Stein discrepancies in (3) can be specified by KSD (as defined by $\mathcal{S}_k$ in (11)) or general Stein discrepancy with an extra critic (as defined by $\mathcal{S}$ in (1)). Hence, the two specifications for $\mathcal{D}_1$ and the two for $\mathcal{D}_2$ ($\mathcal{D}_3$) compose four different combinations in total, and we organize the objectives in each case in Table 4.

In our experiments, we use KSD with RBF kernels for $\mathcal{D}_2$ and $\mathcal{D}_3$ in Joint-W and Joint-JS on two synthetic datasets. For MNIST with conditional training (given the digit class as model input), we also use KSD with RBF kernels. For MNIST and CIFAR with unconditional training (the class is not given as known information), we find that KSD cannot provide desirable results so we adopt general Stein discrepancy for higher model capacity.

The objectives in Table 4 appear to be comutationally expensive. In the worst case (using general Stein discrepancy), there are two minimax operations where one is from GAN or WGAN and one is from Stein discrepancy estimation. To guarantee training efficiency, we alternatively update the generator, estimator, Wasserstein critic and Stein critic over the parameters $\theta$, $\phi$, $\psi$ and $\pi$ respectively. Specifically, in one iteration, we optimize the generator over $\theta$ and the estimator over $\phi$ with one step respectively, and then optimize the Wasserstein critic over $\psi$ with $n_d$ steps and the Stein critic over $\pi$ with $n_c$ steps. Such training approach guarantees the same time complexity order of proposed method as that of GAN or WGAN, and the training time for our model can be bounded within constant times the time for training GAN model. In our experiment, we set $n_d = n_c = 5$ and empirically find that our model Stein Bridging would be two times slower than WGAN on average. We present the training algorithm for Stein Bridging in Algorithm 1.

### E.3 IMPLEMENTATION DETAILS

We give the information of network architectures and hyper-parameter settings for our model as well as each competitor in our experiments.

---

**Algorithm 1:** Training Algorithm for Stein Bridging

---

1 **REQUIRE:** observed training samples $\{\mathbf{x}\} \sim \mathbb{P}_{real}$.
2 **REQUIRE:** $\theta_0, \phi_0, \psi_0, \pi_0$, initial parameters for generator, estimator, Wasserstein critic and Stein critic models respectively. $\alpha^E = 0.0002, \beta_1^E = 0.9, \beta_2^E = 0.999$, Adam hyper-parameters for explicit models. $\alpha^I = 0.0002, \beta_1^I = 0.5, \beta_2^I = 0.999$, Adam hyper-parameters for implicit models. $\lambda_1 = 1, \lambda_2$, weights for $\mathcal{D}_2$ and $\mathcal{D}_3$ (we suggest increasing $\lambda_2$ from 0 to 1 through training). $n_d = 5, n_c = 5$ number of iterations for Wasserstein critic and Stein critic, respectively, before one iteration for generator and estimator. $B = 100$, batch size.
3 **while** *not converged* **do**
4      **for** $n = 1, \cdots, n_d$ **do**
5          Sample $B$ true samples $\{\mathbf{x}_i\}_{i=1}^B$ from $\{\mathbf{x}\}$;
6          Sample $B$ random noise $\{\mathbf{z}_i\}_{i=1}^B \sim P_0$ and obtain generated samples $\widetilde{\mathbf{x}}_\mathbf{i} = G_\theta(\mathbf{z}_i)$ ;
7          $\mathcal{L}_{dis} = \frac{1}{B}\sum_{i=1}^B d_\psi(\mathbf{x}_i) - d_\psi(\widetilde{\mathbf{x}}_\mathbf{i}) - \lambda(\|\nabla_{\hat{\mathbf{x}}_\mathbf{i}} d_\psi(\hat{\mathbf{x}}_\mathbf{i})\| - 1)^2$ // the last term is for gradient penalty in WGAN-GP where $\hat{\mathbf{x}}_\mathbf{i} = \epsilon_i\mathbf{x}_i + (1 - \epsilon_i)\widetilde{\mathbf{x}}_\mathbf{i}, \epsilon_i \sim U(0,1)$;
8          $\psi_{k+1} \leftarrow Adam(-\mathcal{L}_{dis}, \psi_k, \alpha^I, \beta_1^I, \beta_2^I)$// update the Wasserstein critic;
9      **for** $n = 1, \cdots, n_c$ **do**
10          Sample $B$ true samples $\{\mathbf{x}_i\}_{i=1}^B$ from $\{\mathbf{x}\}$;
11          Sample $B$ random noise $\{\mathbf{z}_i\}_{i=1}^B \sim P_0$ and obtain generated samples $\widetilde{\mathbf{x}}_\mathbf{i} = G_\theta(\mathbf{z}_i)$ ;
12          $\mathcal{L}_{critic} = \frac{1}{B}\sum_{i=1}^B \lambda_1 \mathcal{A}_{p_\phi}[\mathbf{f}_\pi(\mathbf{x})] + \lambda_2 \mathcal{A}_{p_\phi}[\mathbf{f}_\pi(\widetilde{\mathbf{x}}_\mathbf{i})]$;
13          $\pi_{k+1} \leftarrow Adam(-\mathcal{L}_{critic}, \pi_k, \alpha^E, \beta_1^E, \beta_2^E)$// update the Stein critic;
14      Sample $B$ random noise $\{\mathbf{z}_i\}_{i=1}^B \sim P_0$ and obtain generated samples $\widetilde{\mathbf{x}}_\mathbf{i} = G_\theta(\mathbf{z}_i)$ ;
15      $\mathcal{L}_{est} = \frac{1}{B}\sum_{i=1}^B \lambda_1 \mathcal{A}_{p_\phi}[\mathbf{f}_\pi(\mathbf{x})] + \lambda_2 \mathcal{A}_{p_\phi}[\mathbf{f}_\pi(\widetilde{\mathbf{x}}_\mathbf{i})]$;
16      $\phi_{k+1} \leftarrow Adam(\mathcal{L}_{est}, \phi_k, \alpha^E, \beta_1^E, \beta_2^E)$// update the density estimator;
17      $\mathcal{L}_{gen} = \frac{1}{B}\sum_{i=1}^B -d_\psi(\widetilde{\mathbf{x}}_\mathbf{i}) + \lambda_2 \mathcal{A}_{p_\phi}[\mathbf{f}_\pi(\widetilde{\mathbf{x}}_\mathbf{i})]$;
18      $\theta_{k+1} \leftarrow Adam(\mathcal{L}_{gen}, \theta_k, \alpha^I, \beta_1^I, \beta_2^I)$// update the sample generator;
19 **OUTPUT:** trained sample generator $G_\theta(\mathbf{z})$ and density estimator $p_\phi(\mathbf{x})$.

---

The energy function is often parametrized as a sum of multiple experts (Hinton (1999)) and each expert can have various function forms depending on the distributions. If using sigmoid distribution, the energy function becomes (see section 2.1 in Kim & Bengio (2017) for details)

$$E_\phi(\mathbf{x}) = \sum_i \log(1 + e^{-(\mathbf{W}_i n(\mathbf{x}) + b_i)}), \tag{31}$$

where $n(\mathbf{x})$ maps input $\mathbf{x}$ to a feature vector and could be specified as a deep neural network, which corresponds to deep energy model (Ngiam et al. (2011))

For synthetic datasets, we set the noise dimension as 4. All the generators are specified as a three-layer fully-connected (FC) neural network with neuron size $4 - 128 - 128 - 2$, and all the Wasserstein critics (or the discriminators in JS-divergence-based GAN) are also a three-layer FC network with neuron size $2 - 128 - 128 - 1$. For the estimators, we set the expert number as 4 and the feature function $n(\mathbf{x})$ is a FC network with neuron size $2 - 128 - 128 - 4$. Then in the last layer we sum the outputs from each expert as the energy value $E(\mathbf{x})$. The activation units are searched within $[LeakyReLU, tanh, sigmoid, softplus]$. The learning rate $[1e-6, 1e-5, 1e-4, 1e-3, 1e-2]$ and the batch size $[50, 100, 150, 200]$. The gradient penalty weight for WGAN is searched in $[0, 0.1, 1, 10, 100]$.

For MNIST dataset, we set the noise dimension as 100. All the critics/discriminators are implemented as a four-layer network where the first two layers adopt convolution operations with filter size 5 and stride $[2, 2]$ and the last two layers are FC layers. The size for each layer is $1 - 64 - 128 - 256 - 1$. All the generators are implemented as a four-layer networks where the first two layers are FC and the last two adopt deconvolution operations with filter size 5 and stride $[2, 2]$. The size for each layer is $100 - 256 - 128 - 64 - 1$. For the estimators, we consider the expert number as 128 and the feature function is the same as the Wasserstein critic except that the size of last layer is 128. Then we sum the outputs from each expert as the energy

| Method | Two-Cirlce | | | | | | Two-Spiral | | | | |
|---|---|---|---|---|---|---|---|---|---|---|---|
| | MMD | HSR | KLD | JSD | AUC | | MMD | HSR | KLD | JSD | AUC |
| GAN | 0.0033 | 0.772 | - | - | - | | 0.0082 | 0.583 | - | - | - |
| GAN+LR | 0.0106 | 0.391 | - | - | - | | 0.0068 | 0.821 | - | - | - |
| GAN+ER | 0.0103 | 0.428 | - | - | - | | 0.0071 | 0.780 | - | - | - |
| GAN+VA | 0.0118 | 0.295 | - | - | - | | 0.0085 | 0.761 | - | - | - |
| WGAN-GP | 0.0010 | 0.841 | - | - | - | | 0.0090 | 0.697 | - | - | - |
| WGAN+LR | 0.0013 | 0.840 | - | - | - | | 0.0095 | 0.607 | - | - | - |
| WGAN+ER | 0.0008 | 0.830 | - | - | - | | 0.0182 | 0.730 | - | - | - |
| WGAN+VA | 0.0016 | 0.835 | - | - | - | | 0.0159 | 0.618 | - | - | - |
| DEM | - | - | 2.036 | 0.431 | 0.683 | | - | - | 1.206 | 0.315 | 0.640 |
| EGAN | - | - | 3.350 | 0.474 | 0.616 | | - | - | 1.916 | 0.445 | 0.499 |
| DGM | 0.0040 | 0.774 | 2.272 | 0.445 | 0.600 | | 0.0019 | 0.833 | 1.725 | 0.414 | 0.589 |
| Joint-JS | 0.0037 | **0.883** | 1.104 | 0.297 | **0.962** | | 0.0031 | 0.717 | 0.655 | 0.193 | 0.808 |
| Joint-W | **0.0007** | 0.844 | **1.030** | **0.281** | 0.961 | | **0.0003** | **0.909** | **0.364** | **0.110** | **0.810** |

Table 5: Quantitative results including MMD (lower is better), HSR (higher is better) as the metrics for quality of generated samples and KLD (lower is better), JSD (lower is better), AUC (higher is better) as the metrics for accuracy of estimated densities on Two-Circle and Two-Spiral datasets.

value. The activation units are searched within $[ReLU, LeakyReLU, tanh]$. The learning rate $[2e-5, 2e-4, 2e-3, 2e-2]$ and the batch size $[32, 64, 100, 128]$. The gradient penalty weight for WGAN is searched in $[1, 10, 100, 1000]$.

For CIFAR dataset, we adopt the same architecture as DCGAN for critics and generators. As for the estimator, the architecture of feature function is the same as the critics except the last year where we set the expert number as 128 and sum each output as the output energy value. The architectures for Stein critic are the same as Wasserstein critic for both MNIST and CIFAR datasets. In other words, we consider $d' = 1$ in (1) and further simply $\phi$ as an average of each dimension of $\mathbb{E}_{\mathbf{x} \sim \mathbb{P}}[\mathcal{A}_{\mathbb{Q}}\mathbf{f}(\mathbf{x})]$. Empirically we found this setting can provide efficient computation and decent performance.

E.4   EVALUATION METRICS

We adopt some quantitative metrics to evaluate the performance of each method on different tasks. In section 4.1, we use two metrics to test the sample quality: Maximum Mean Discrepancy (MMD) and High-quality Sample Rate (HSR). MMD measures the discrepancy between two distributions $X$ and $Y$, $MMD(X, Y) = \|\frac{1}{n} \sum_{i=1}^{n} \Phi(x_i) - \frac{1}{m} \sum_{j=1}^{m} \Phi(y_i)\|$ where $x_i$ and $y_j$ denote samples from $X$ and $Y$ respectively and $\Phi$ maps each sample to a RKHS. Here we use RBF kernel and calculate MMD between generated samples and true samples. HSR statistics the rate of high-quality samples over all generated samples. For Two-Cirlce dataset, we define the generated points whose distance from the nearest Gaussian component is less than $\sigma_1$ as high-quality samples. We generate 2000 points in total and statistic HSR. For Two-Spiral dataset, we set the distance threshold as $5\sigma_2$ and generate 5000 points to calculate HSR.

As for Inception Score and CEPC. For MNIST, we pre-train a classifier for 10 digits which can provide the test accuracy up to $99\%$ for calculation of scores. The conditional entropy of predicted classes (CEPC) for given samples is defined as $\mathbb{H}(y|x) \approx \frac{1}{n} \sum_{i=1}^{n} \sum_{k=1}^{10} p(y_k|x_i) \log p(y_k|x_i)$ where $x$ is a generated instance and $y$ denotes the predicted class given $x$ from a pre-trained classifier. CEPC measures how well a given sample can be classfied into a right class, i.e. the quality of such sample. For CIFAR, we use the Inception V3 Network in Tensorflow as pre-trained classifier.

In section 4.2, we use three metrics to characterize the performance for density estimation: KL divergence, JS divergence and AUC. We divide the map into a 300 meshgrid, calculate the unnormalized density values of each point given by the estimators and compute the KL and JS divergences between estimated density and ground-truth density. Besides, we select the centers of each Gaussian components as positive examples (expected to have high densities) and randomly sample 10 points within a circle around each center as negative examples (expected to have relatively low densities) and rank them according to the densities given by the model. Then we obtain the area under the curve (AUC) for false-positive rate v.s. true-positive rate.

| Class | | '0' | '1' | '2' | '3' | '4' | '5' | '6' | '7' | '8' | '9' |
|---|---|---|---|---|---|---|---|---|---|---|---|
| WGAN-GP | $l_1$ | 20.3 | 11.4 | 14.3 | 14.8 | 13.5 | 13.3 | 13.8 | 11.0 | 13.0 | 12.3 |
| | $l_2$ | 1.74 | 1.07 | 0.82 | 0.98 | 0.83 | 0.68 | 0.95 | 0.62 | 0.82 | 0.75 |
| WGAN+LR | $l_1$ | 13.8 | 5.9 | 13.6 | 19.1 | 11.8 | 18.3 | 10.7 | 11.5 | 14.0 | 9.9 |
| | $l_2$ | 0.80 | 0.34 | 0.84 | 1.81 | 0.65 | 1.37 | 0.62 | 0.70 | 0.90 | 0.57 |
| WGAN+ER | $l_1$ | 16.1 | 8.9 | 11.7 | 14.2 | 12.3 | 10.8 | 13.9 | 11.4 | 12.1 | 10.9 |
| | $l_2$ | 1.20 | 0.74 | 0.54 | 0.86 | 0.73 | 0.54 | 0.97 | 0.69 | 0.72 | 0.63 |
| WGAN+VA | $l_1$ | 16.3 | 7.1 | 13.7 | 13.7 | 11.9 | 13.2 | 13.6 | 11.2 | 12.1 | 10.6 |
| | $l_2$ | 1.12 | 0.35 | 0.81 | 0.85 | 0.69 | 0.76 | 1.04 | 0.71 | 0.74 | 0.71 |
| DGM | $l_1$ | 22.2 | 10.9 | 12.7 | 10.2 | 10.8 | 9.0 | 9.5 | 10.9 | 12.7 | 11.7 |
| | $l_2$ | 1.41 | 0.83 | 0.81 | 0.65 | 0.67 | 0.56 | 0.66 | 0.67 | 0.88 | 0.76 |
| Joint-W | $l_1$ | 14.1 | 7.5 | 14.3 | 12.9 | 11.1 | 11.0 | 13.7 | 9.7 | 12.0 | 11.5 |
| | $l_2$ | 0.89 | 0.47 | 0.93 | 0.73 | 0.55 | 0.51 | 1.06 | 0.53 | 0.70 | 0.97 |

Table 6: $l1$ and $l2$ distances between means of true digits and generated digits in each class on MNIST.

| Class | | '0' | '1' | '2' | '3' | '4' | '5' | '6' | '7' | '8' | '9' |
|---|---|---|---|---|---|---|---|---|---|---|---|
| WGAN-GP | $l_1$ | 80.8 | 82.7 | 40.2 | 69.3 | 44.7 | 59.2 | 77.6 | 107.7 | 50.81 | 89.3 |
| | $l_2$ | 1.75 | 1.84 | 0.92 | 1.57 | 1.04 | 1.40 | 1.78 | 2.32 | 1.78 | 1.92 |
| WGAN+LR | $l_1$ | 78.4 | 79.2 | 73.8 | 86.0 | 75.8 | 77.2 | 106.7 | 103.0 | 56.5 | 92.3 |
| | $l_2$ | 1.63 | 1.76 | 1.59 | 1.88 | 1.68 | 1.74 | 2.36 | 2.23 | 1.24 | 2.00 |
| WGAN+ER | $l_1$ | 75.5 | 64.0 | 100.0 | 65.0 | 58.5 | 69.1 | 74.5 | 81.8 | 62.5 | 71.3 |
| | $l_2$ | 1.56 | 1.45 | 2.04 | 1.43 | 1.35 | 1.57 | 1.67 | 1.82 | 1.40 | 1.58 |
| WGAN+VA | $l_1$ | 60.9 | 70.0 | 79.4 | 62.7 | 63.0 | 73.9 | 76.2 | 77.2 | 59.8 | 66.4 |
| | $l_2$ | 1.32 | 1.55 | 1.68 | 1.39 | 1.42 | 1.63 | 1.70 | 1.77 | 1.33 | 1.48 |
| DGM | $l_1$ | 167.8 | 185.0 | 149.4 | 250.1 | 105.3 | 134.0 | 223.8 | 197.3 | 148.3 | 231.7 |
| | $l_2$ | 3.67 | 4.14 | 3.15 | 5.41 | 2.39 | 3.04 | 4.68 | 4.51 | 3.24 | 5.25 |
| Joint-W | $l_1$ | 59.3 | 58.1 | 77.3 | 54.8 | 58.1 | 65.1 | 63.9 | 82.8 | 59.1 | 63.2 |
| | $l_2$ | 1.26 | 1.30 | 1.60 | 1.23 | 1.28 | 1.44 | 1.44 | 1.80 | 1.27 | 1.43 |

Table 7: $l1$ and $l2$ distances between means of true images and generated images in each class on CIFAR. (Class '0', '1', '2', '3', '4', '5', '6', '7', '8' and '9' stand for 'airplane', 'automobile', 'bird', 'cat', 'deer', 'dog', 'frog', 'horse', 'ship' and 'truck' respectively.)

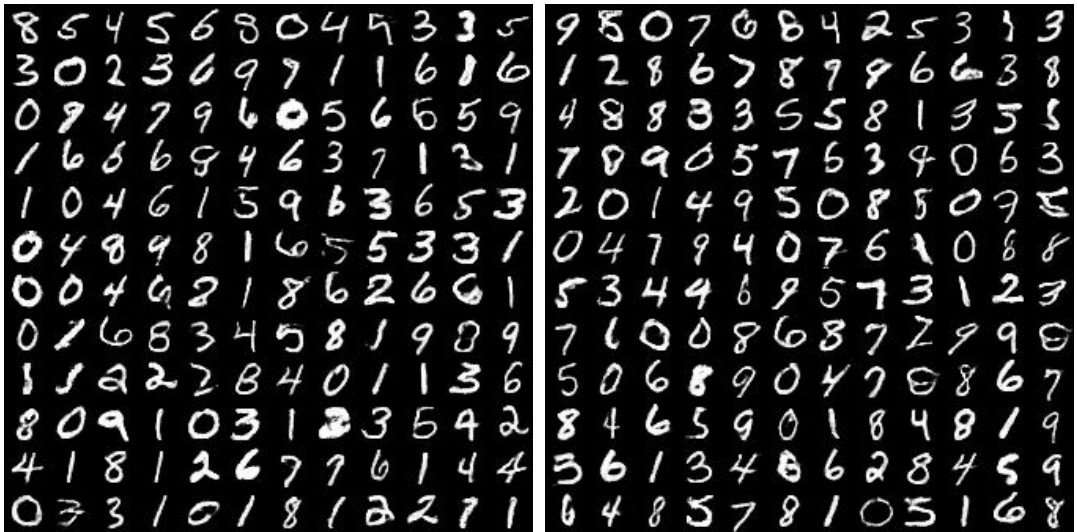

(a) Randomly sampled over all digits

(b) Randomly sampled over digits with top 50% densities

Figure 9: Generated digits given by Joint-W on MNIST.

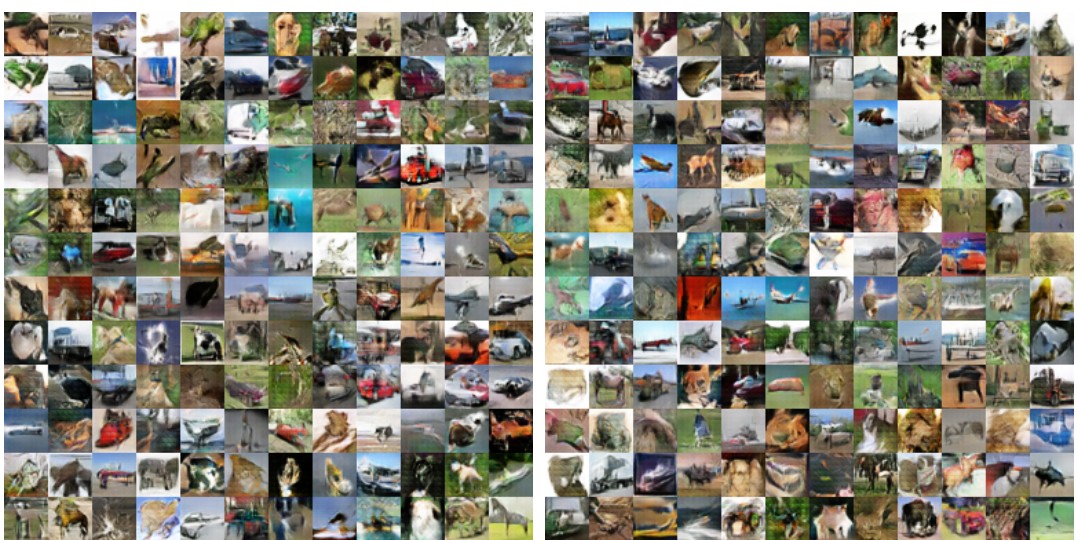

(a) Randomly sampled over all images

(b) Randomly sampled over images with top 50% densities

Figure 10: Generated images given by Joint-W on CIFAR.

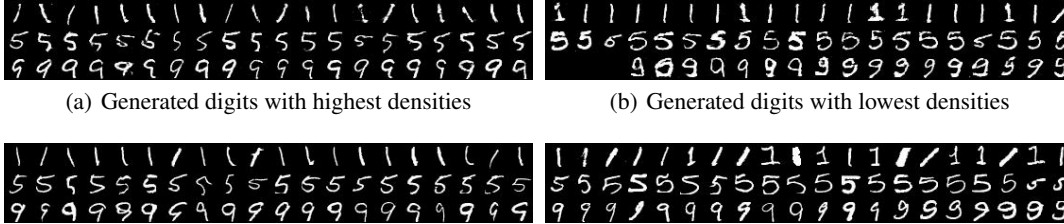

(a) Generated digits with highest densities

(b) Generated digits with lowest densities

(c) Real digits with highest densities

(d) Real digits with lowest densities

Figure 11: The generated digits (and real digits) with the highest densities and the lowest densities given by Joint-W.

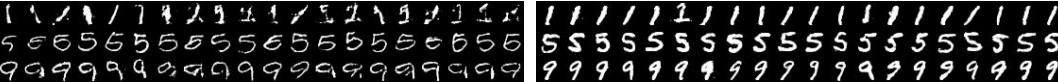

(a) Generated digits with highest densities      (b) Generated digits with lowest densities

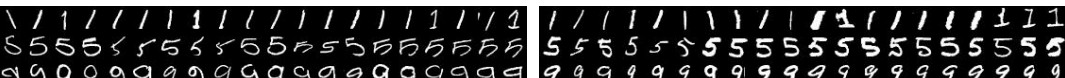

(c) Real digits with highest densities      (d) Real digits with lowest densities

Figure 12: The generated digits (and real digits) with the highest densities and the lowest densities given by DGM.

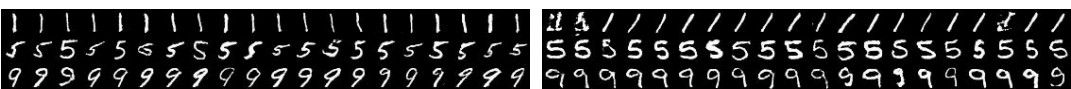

(a) Generated digits with highest densities      (b) Generated digits with lowest densities

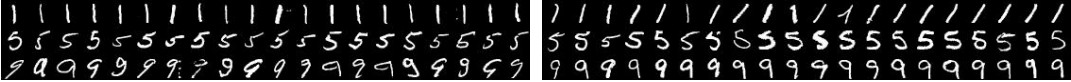

(c) Real digits with highest densities      (d) Real digits with lowest densities

Figure 13: The generated digits (and real digits) with the highest densities and the lowest densities given by EGAN.

