# OpenReview forum: "Stein Bridging: Enabling Mutual Reinforcement between Explicit and Implicit Generative Models"
_ICLR.cc/2020/Conference — Reject_

### Official Review · AnonReviewer2 · 2019-10-22
**Official Blind Review #2**

**Rating:** 1

**Review:**

This paper proposes a training objective that combines three terms:
* A Stein discrepancy for learning a energy model with intractable normalizing constant
* A Wasserstein GAN objective for learning an implicit neural sampler.
* A Stein discrepancy for minimizing the distance between distributions defined by the energy model and the GAN.

The third term is called "Stein bridging" by the authors. It seems pretty difficult to motivate such a bridging term because from the first glance this term does not add anything to learning the two models from data. So I'm wondering how the authors motivate themselves to study this modification. This is my main concern about the paper if this term appears simply because that energy model has an unnormalized density while from GANs we can sample, and Stein discrepancy is best applicable to such pair of distributions.

Apart from the concern on motivations, I tried to follow the arguments in Section 3, as the bridging term is justified as regularization to both models. However, I think the proof of Theorem 1 is incorrect:

* I don't think it makes sense from \nabla \log (1 + h(x)) to (1 + h(x))\nabla h(x), even if the taylor expansion suggested by the authors is applied.
* In the next step, "Consider a further approximation", this approximation basically sets 1 + h(x) to 1, if h(x) is approximately 0, then P=P_E..

Minor:
* IN proof of Theorem 1, the expectation should be always over x,x'~P_E instead of x~P, right?




**Experience Assessment:**

I have published one or two papers in this area.

**Review Assessment: Checking Correctness Of Derivations And Theory:**

I carefully checked the derivations and theory.

**Review Assessment: Checking Correctness Of Experiments:**

I assessed the sensibility of the experiments.

**Review Assessment: Thoroughness In Paper Reading:**

I read the paper at least twice and used my best judgement in assessing the paper.

---

> ### Author Response · Authors · 2019-11-14
> **Response to Reviewer #2**
>
> We appreciate your comments, and we apologize for the typos that cause some confusion.
>
>
> 1. To begin with, we apologize for insufficient illumination of the motivation why we consider a Stein discrepancy as a bridge between two models. Here we provide a more thorough discussions from three perspectives.
>
> Firstly, there are many of applications where we do need both of the explicit density (at least an energy value that can distinguish high-quality samples and low-quality ones) and sample generation. In the introduction part, we discuss some of them, like sample evaluation, data augmentation for insufficient observation and outlier detection. In our experiments, we apply our model to address data insufficiency and outlier detection.
> We also observe in the literature of GAN, quite a bit is devoted to the discussion of estimating the likelihood for the obtained generator.
>
> Secondly, jointly training two models can presumably compensate and reinforce each other in the training process. Although the ideal global optimum of an individual explicit or implicit model can both guarantee that the model exactly captures the data distribution, when it comes to practical training, an individually learned model could suffer from many issues like mode collapse and training unstability that could lead to undesirable performance. Hence, one important motivation of joint training is to let one model regularize the other and help it avoid the local optima or stablize the training. We verify these arguments in Section 3.2 and 4.
>
> Thirdly, in some specific tasks, we need to add some induction bias to the model but it is often the case that it is easy for one model to incorporate the induction bias while it is harder for another. For example, if we want to obtain a certain type of generated samples, then it is difficult to mathematically enforce some constraints on the implicit model, but we can consider a truncated density/energy function for explicit model. In this case, the explicit model can guide the implicit one to generate the samples that meets the requirements through joint training. If we specify energy model as PixelCNN, it would be easy to add induction bias that could control pixel-level features of generated images. In fact, we do some additional experiments where we replace the original deep energy model as PixelCNN++, and we achieve better results of generated samples with inception score 7.20.
>
> 2. The approximations in the first version are not necessary and we apologize for the confusion due to some typos. We modify this part in the updated version as below:
>
> \[
> \begin{aligned}
> & \min_{h:\mathbb{E}_{\mathbb{P}_E}[h]=0} \{\mathbb{E}_{\mathbb{P}_E}[hD] + \lambda_2\cdot \mathbb{E}_{\mathbf{x},\mathbf{x}'\sim\mathbb{P}}[\nabla_x\log(1+h(\mathbf{x}))^\top k(\mathbf{x},\mathbf{x}') \nabla_x\log(1+h(\mathbf{x}'))] \}\\
> = & \min_{h:\mathbb{E}_{\mathbb{P}_E}[h]=0} \left\{\mathbb{E}_{\mathbb{P}_E}[hD] + \lambda_2\cdot \mathbb{E}_{\mathbf{x},\mathbf{x}'\sim\mathbb{P}}\left[\frac{\nabla_x h(\mathbf{x})^\top}{1+h(\mathbf{x})} k(\mathbf{x},\mathbf{x}') \frac{\nabla_x h(\mathbf{x}')}{1+h(\mathbf{x}')}\right] \right\}\\
> = & \min_{h:\mathbb{E}_{\mathbb{P}_E}[h]=0} \left\{\mathbb{E}_{\mathbb{P}_E}[hD] + \lambda_2\cdot \mathbb{E}_{\mathbf{x},\mathbf{x}'\sim\mathbb{P}_E}\left[\nabla_x h(\mathbf{x})^\top k(\mathbf{x},\mathbf{x}') \nabla_x h(\mathbf{x}')\right] \right\}.
> \end{aligned}
> \]
>
> 3. Yes, the expectation should be over $\mathbf{x},\mathbf{x}'\sim\mathbb{P}_E$. Thanks for pointing this out.

---

> > ### Comment · AnonReviewer2 · 2019-11-14
> > **can't see how the proof works**
> >
> > Hi,
> >
> > Can you explain how this line works?
> >
> > \begin{aligned}
> > \min_{h:\mathbb{E}_{\mathbb{P}_E}[h]=0} \left\{\mathbb{E}_{\mathbb{P}_E}[hD] + \lambda_2\cdot \mathbb{E}_{\mathbf{x},\mathbf{x}'\sim\mathbb{P}}\left[\frac{\nabla_x h(\mathbf{x})^\top}{1+h(\mathbf{x})} k(\mathbf{x},\mathbf{x}') \frac{\nabla_x h(\mathbf{x}')}{1+h(\mathbf{x}')}\right] \right\}\\
> > = & \min_{h:\mathbb{E}_{\mathbb{P}_E}[h]=0} \left\{\mathbb{E}_{\mathbb{P}_E}[hD] + \lambda_2\cdot \mathbb{E}_{\mathbf{x},\mathbf{x}'\sim\mathbb{P}_E}\left[\nabla_x h(\mathbf{x})^\top k(\mathbf{x},\mathbf{x}') \nabla_x h(\mathbf{x}')\right] \right\}.
> > \end{aligned}

---

> > > ### Author Response · Authors · 2019-11-14
> > > **This is a consequence of the definition of the density ratio**
> > >
> > > According to the definition $h=d\mathbb{P}/d\mathbb{P}_E-1$, we have $d\mathbb{P}=(1+h)d\mathbb{P}_E$. Replacing $d\mathbb{P}$ with $(1+h)d\mathbb{P}_E$ on the left side yields the right side.

---

> > > > ### Comment · AnonReviewer2 · 2019-11-14
> > > > **following**
> > > >
> > > > What do you mean by min ... + \lambda r^2 : E[...] <= r^2 in the derivation following?

---

> > > > > ### Author Response · Authors · 2019-11-14
> > > > > **Treat $r^2$ as an auxiliary for $ \mathbb{E}_{\mathbf{x},\mathbf{x}'\sim\mathbb{P}_E}\left[\nabla_x h(\mathbf{x})^\top k(\mathbf{x},\mathbf{x}') \nabla_x h(\mathbf{x}')\right]$**
> > > > >
> > > > > Basically, we introduce an auxiliary variable $r^2$ to represent the value of $ \mathbb{E}_{\mathbf{x},\mathbf{x}'\sim\mathbb{P}_E}\left[\nabla_x h(\mathbf{x})^\top k(\mathbf{x},\mathbf{x}') \nabla_x h(\mathbf{x}')\right]$. The minimization over $r$ and the inequality constraint $ \mathbb{E}_{\mathbf{x},\mathbf{x}'\sim\mathbb{P}_E}\left[\nabla_x h(\mathbf{x})^\top k(\mathbf{x},\mathbf{x}') \nabla_x h(\mathbf{x}')\right] \leq r^2$ forces $r^2=\mathbb{E}_{\mathbf{x},\mathbf{x}'\sim\mathbb{P}_E}\left[\nabla_x h(\mathbf{x})^\top k(\mathbf{x},\mathbf{x}') \nabla_x h(\mathbf{x}')\right]$.

---

### Official Review · AnonReviewer3 · 2019-10-22
**Official Blind Review #3**

**Rating:** 3

**Review:**

This paper proposes to train a GAN and an EBM jointly, and bridge them using a Stein discrepancy. The paper claims it leads to novel regularization effect on both models, and stablizes the optimization process. Experiments on MNIST and CIFAR-10 show improvement in sample quality and outlier detection.

Both the idea and the experiment results are interesting. However, the derivations contain too many typos and are in general confusing, and I cannot confirm their correctness. Therefore I cannot recommend acceptance.

Specifically the proof of Theorem 1 seems problematic:
1. In the proof you claim (15) equals $\frac{-1}{4\lambda_2}\lVert D-t\rVert_{H^{-1}}$. But (15) could only simplify to
$\frac{1}{\lambda_2} ( E[D\cdot(\lambda_2 h)] + E_{x,x'}[\nabla(\lambda_2 h(x))^T k(x,x') \nabla(\lambda_2h(x'))],$
where h is unconstrained. Compare this with the definition of the $H^{-1}$ norm,
$sup_h \{E[D\cdot h]: E_{x,x'}[\nabla h(x)^T k(x,x') \nabla h(x')] \le 1\},$
how did you drop the inequality constraint on h?
2. The transformation from the original objective (14) to (15) is strange as well. In the proof you claim the minimization problem below "invoking Lagrangian duality gives" could only turn to (15) after "applying the approximation log(1+a)=a+O(a^2)" and "a further approximation". But you can turn it into
E[(D-t)h]+λ E_{x,x'~P_E}[∇h(x)^T k(x,x') ∇h(x')]
simply by simplifying the gradient terms. Also, why did the $t$ disappear in (15)?

There are also typos and issues elsewhere. To list a few:
3. Energy-based models are not generally referred to as "explicit models", since the normalization constant is intractable. I would suggest to replace the occurrences of (log) "density" with "energy" to avoid confusion.
4. The GD update rule of (6) is incorrect; the optima should also be (0, 1), instead of (1, 0).
5. On the second line on Page 8, the unnormalized log density cannot be x^2+\phi x, as the normalization constant would then be infinity.

For these reasons, I believe this paper needs a thorough proofreading before it can be reviewed efficiently.

**Experience Assessment:**

I have read many papers in this area.

**Review Assessment: Checking Correctness Of Derivations And Theory:**

I assessed the sensibility of the derivations and theory.

**Review Assessment: Checking Correctness Of Experiments:**

I assessed the sensibility of the experiments.

**Review Assessment: Thoroughness In Paper Reading:**

I read the paper at least twice and used my best judgement in assessing the paper.

---

> ### Author Response · Authors · 2019-11-14
> **Response to Reviewer #3**
>
> We appreciate very much your helpful comments and apologize for the typos that cause some confusion.
>
> 1. We add the following reasoning in the updated version:
>
> \[\begin{aligned}
>     &\min_{h:\mathbb{E}_{\mathbb{P}_E}[h]=0} \left\{\mathbb{E}_{\mathbb{P}_E}[hD] + \lambda_2\cdot \mathbb{E}_{\mathbf{x},\mathbf{x}'\sim\mathbb{P}}\left[\nabla_x h(\mathbf{x})^\top k(\mathbf{x},\mathbf{x}') \nabla_x h(\mathbf{x}')\right] \right\} \\
>     = & \min_{r\ge0} \min_{h:\mathbb{E}_{\mathbb{P}_E}[h]=0} \left\{\mathbb{E}_{\mathbb{P}_E}[hD] + \lambda_2 r^2:  \mathbb{E}_{\mathbf{x},\mathbf{x}'\sim\mathbb{P}}\left[\nabla_x h(\mathbf{x})^\top k(\mathbf{x},\mathbf{x}') \nabla_x h(\mathbf{x}')\right] \leq r^2 \right\} \\
>     = & \min_{r\ge0} \min_{h:\mathbb{E}_{\mathbb{P}_E}[h]=0} \left\{r\mathbb{E}_{\mathbb{P}_E}[hD] + \lambda_2 r^2:  \mathbb{E}_{\mathbf{x},\mathbf{x}'\sim\mathbb{P}}\left[\nabla_x h(\mathbf{x})^\top k(\mathbf{x},\mathbf{x}') \nabla_x h(\mathbf{x}')\right] \leq 1 \right\}\\
>     = & \min_{r\ge0} \ \left\{\lambda_2 r^2 - r||D||_{H^{-1}(\mathbb{P}_E;k)} \right\}\\
>     = & -\frac{1}{4\lambda_2} ||D||_{H^{-1}(\mathbb{P}_E;k)}.
>     \end{aligned}
>   \]
>   Note that we slightly change the definition of the kernel Sobolev dual norm just to get a cleaner result:
>   \[
> 	||D||_{H^{-1}(\mathbb{P};k)} := \sup_{u\in C_0^\infty}\left\{\langle D,u \rangle_{L^2(\mathbb{P})}:\ \mathbb{E}_{\mathbf{x},\mathbf{x}'\sim\mathbb{P}}\left[\nabla_x h(\mathbf{x})^\top k(\mathbf{x},\mathbf{x}') \nabla_x h(\mathbf{x}')\right] \leq 1,\  \mathbb{E}_{\mathbb{P}}[h]=0\right\}.
> 	\]
>
> 2. As you pointed out, the approximation is indeed unnecessary, and we modify the proof (see the second bullet in response to reviewer 2). $t$ should appear in (15) -- we apologize again for the typo and as above-mentioned, in the updated version, we slightly change the definition of the kernel Sobolev dual norm just to get a cleaner result without explicitly having $t$.
>
> 3. Thanks for pointing out the confusing terminology. We have changed all `density' into `log-density' or `energy'.
>
> 4. We have modified the update rule according to your suggesion in the updated version.
>
> 5. We have changed the assumed energy model to $p(x)=\exp(-\frac{1}{2}x^2-\phi x)$ to avoid the infinite normalizing constant, and the results remain the same.

---

### Official Review · AnonReviewer1 · 2019-10-23
**Official Blind Review #1**

**Rating:** 3

**Review:**

Thank you for your rebuttal. The paper improved after the rebuttal but I still think point 5 in the rebuttal is problematic since using d'=d, may not guarantee that we have a proper metric as claimed by the authors. I m updating  my score as a weak reject for this paper.

Summary of the paper:

The paper proposes to train implicit  model such as gan and an explicit model (Energy Based ) jointly . The GAN is trained using WGAN-GP objective or the original JS objective (we have a discriminator D and Generator G). The energy based model (E) is trained using Stein Divergence with a fixed kernel k or a learned critic who's parameters are denoted pi in the paper. Note that the critic of the stein divergence is vector valued. This paper propose to add a regularization loss on the stein divergence between the generator G (implicit model ) and the explicit model (E). This gives a training objective

$\min_{G,E} W(P_r, G) + \lambda_1 S(P_r, P_E)+ \lambda_2 S(P_{G}, P_{E})$

In the paper the stein critic is shared between the two stein divergence which means that the authors are rather considering : $S(\lambda_1 P_r + \lambda_ 2P_{G}, P_{E})$

Paper shows the effect of this additional coupling between the two models as a regularization on the Discriminator D and on the critic of the stein divergence.

Then  the effect of the regularization is also show in terms of convergence in the optimization on a bilinear game, and in the convex concave case.

Experiments are given showing the benefits of the joint training.

Review:

The paper has a lot of typos and needs a lot of proofreading and is not in shape for being reviewed.
There are too many concerns with this papers:

1- The first one was mentioned above if the critic is shared then you better be considering :  $S(\lambda_1 P_r + \lambda_ 2P_{G}, P_{E})$

2- In equation 4,  the problem is $\min_{G} \max_{D}$ it is swapped.

3- There a lot of gaps in the proofs of Theorems 1 and 2. The transition from equation 14 to the $\inf_{\mathbb{P}}...$ is not explained and seems flawed. In theorem 2 , the proof is too short and swapping of $\min$ and $\mathbb{E}$ is not backed rigoursly.

4- Again in Equation 8, it is not clear how the Stein terms were computed , the appendix does not give the derivations either.

5 - Authors say that the Stein critic have similar architecture to the GAN critic , which indicates an error in the implementation in the neural case for stein critic. Stein critic has to be vector valued, after checking the code of this paper on GitHub, stein critic maps to a real value in the code , which is flawed. The critic of stein needs to map the image to an image , which actually quite expensive.

Typos:
abstract : without explicitly defines -> defining
multimodal data . has been -> have
without explicit defines -> defining


**Experience Assessment:**

I have published in this field for several years.

**Review Assessment: Checking Correctness Of Derivations And Theory:**

I carefully checked the derivations and theory.

**Review Assessment: Checking Correctness Of Experiments:**

I carefully checked the experiments.

**Review Assessment: Thoroughness In Paper Reading:**

I read the paper thoroughly.

---

> ### Author Response · Authors · 2019-11-14
> **Response to Review #1**
>
> We appreciate very much your comments and suggestions. We apologize for the typos and unjustified claims in the first version and we correct them in the updated version.
>
> 1. As you kindly mentioned, we can directly consider $S(\lambda_1P_{real}+\lambda_2P_G, P_E)$ when  $\lambda_1+\lambda_2=1$ (after suitable scaling). In fact, in this case
> \[
> S(\lambda_1P_{real}+\lambda_2P_G, P_E) = \lambda_1 S(P_{real}, P_E) + \lambda_2 S(P_G, P_E).
> \]
> Since $\lambda_1P_{real}+\lambda_2P_G$ is a mixture, we can generate samples from it by the usual sampling scheme for mixture models which involve sampling from $P_{real}$ and $P_G$, separately. So theoretically and computationally, the above two approaches make little difference.
>
> One may think to just minimize $S(\lambda_1P_{real}+\lambda_2P_G, P_E)$ (without $W(P_G,P_{real})$). We tried this approach and the numerical results are not good.
>
> There is an advantage of using the $\lambda_1 S(P_{real}, P_E) + \lambda_2 S(P_G, P_E)$ formulation: this is when we actually have a density model for $P_E$ and we want to use MLE for estimating $P_E$ from data, then we can use
> \[
> -\lambda_1 {\cal E}_{P_{real}} [P_E] + \lambda_2 S(P_G, P_E).
> \]
>
> We empirically compared using shared Stein critics and two different Stein critics and found that there was little difference for the performance. So we used shared Stein critics to reduce computational cost.
>
> 2. We add the following reasoning to justify the swap of $\min_G$ and $\max_D$.
>
> Using the notations in the paper, we consider
> \[\begin{aligned}
> \min_{h\in L^1(\mathbb{P}_E):\mathbb{E}_{\mathbb{P}_E}[h]=0} \max_{D:\mathrm{Lip}(D)\leq 1} \bigg\{ \mathbb{E}_{\mathbb{P}_E}[D] + \mathbb{E}_{\mathbb{P}_E}[hD] - \mathbb{E}_{\mathbb{P}_{real}}[D] + \lambda_1\mathcal{S}(\mathbb{P}_{real},\mathbb{P}_E) \\\quad \quad + \lambda_2\cdot \mathbb{E}_{\mathbf{x},\mathbf{x}'\sim\mathbb{P}}[\nabla_x\log(1+h(\mathbf{x}))^\top k(\mathbf{x},\mathbf{x}') \nabla_x\log(1+h(\mathbf{x}'))]\bigg\},
> \end{aligned}
> \]
> where $h(\mathbf{x}) := d\mathbb{P}/d\mathbb{P}_E(\mathbf{x}) -1$ (see next bullet for the validity of this formulation). Without loss of generality, we can only consider those $D$'s with $D(\mathbf{x}_0)=0$ for some element $\mathbf{x}_0$, as a constant shift does not change the value of $\mathbb{E}_{\mathbb{P}_E}[(1+h)D]-\mathbb{E}_{\mathbb{P}_{real}}[D]$.
> The space of Lipschitz functions that vanish at $\mathbf{x}_0$ is a Banach space, and the subset of 1-Lipschtiz functions is compact (Weaver (1999)). Moreover, $L^1(\mathbb{P}_E)$ is also a Banach space. The above verifies the condition of Sion's minimax theorem, and thus the claim is proved.
>
> 3. We apologize for omitting details in the proof of Theorem 1 and 2. We add more explanations in the updated version.
>
> For the transition from (14) to the $\inf_{\mathbb{P}}$ in the proof of Theorem 1, we add the following reasoning:
>
> Assume $\mathbb{P}_G$ exhausts all continuous probability distributions.
> From the definition of kernel Stein discrepancy
> \[
> \mathcal{S}(\mathbb{P},\mathbb{P}_E) = \mathbb{E}_{\mathbf{x},\mathbf{x}'\in\mathbb{P}} [(\nabla_x \log \mathbb{P}(\mathbf{x}) - \nabla_x \log \mathbb{P}_E(\mathbf{x}))^\top k(\mathbf{x},\mathbf{x}') (\nabla_x \log \mathbb{P}(\mathbf{x}') - \nabla_x \log \mathbb{P}_E(\mathbf{x}'))],
> \]
> $\mathcal{S}(\mathbb{P},\mathbb{P}_E)$ is infinite if $\mathbb{P}$ is not absolutely continuous with respect to $\mathbb{P}_E$.
> Hence, it suffices to consider those $\mathbb{P}$'s that are absolutely continuous with respect to $\mathbb{P}_E$.
>
> For the swapping of $\min$ and $\mathbb{E}$ in the proof of Theorem 2, we add the following justification: the exchanging of $\min$ and $\mathbb{E}$ follows from the interchangebability principle (Theorem 7.80 in Shapiro (2009)).
>
> 4. We have supplemented the derivation for (8) (which changes to (6) in the updated version) in Appendix D.1.
>
> 5. We kindly clarify that the Stein critic can be a function $f: R^d\rightarrow R^{d'}$ where $d'$ does not necessarily equal to $d$, see definition 2.1 in Liu et al. (2016). The only requirement for $f$ is Stein identity condition, i.e.,
> $$E_p[A_pf(x)] = E_p[\nabla_x \log p(x) f(x)^\top + \nabla_x f(x)] = 0.$$
> Such property induces a measurement of difference between two distributions $p$ and $q$ as $\mathbb E_p[A_qf(x)]$, which is a $d\times d'$ matrix. The Stein identity guarantees that $ E_p[ A_qf(x)]=0, \forall f$ in some function space, if and only if $p=q$. So a general Stein discrepancy can be written as $\phi( E_p[A_qf(x)])$ where $\phi$ is an operation that transforms a $d\times d'$ matrix into a scalar. A common choice of $\phi$ is trace operation on condition that $d'=d$. Note that one can also consider $d'\neq d$ and use other forms for $\phi$, like matrix norm, as is indicated in Liu et al. (2016). Therefore, for practical implementation we use $d'=1$ and further simply $\phi$ as an average of each dimension of $E_p(A_qf(x))$.

---

### Decision · Program_Chairs · 2019-12-19

**Decision:**

Reject

**Comment:**

The paper proposes a generative model that jointly trains an implicit generative model and an explicit energy based model using Stein's method. There are concerns about technical correctness of the proofs and the authors are advised to look carefully into the points raised by the reviewers.